# What We Are Learning from COVID-19 for Respiratory Protection: Contemporary and Emerging Issues

**DOI:** 10.3390/polym13234165

**Published:** 2021-11-28

**Authors:** Rui Li, Mengying Zhang, Yulin Wu, Peixin Tang, Gang Sun, Liwen Wang, Sumit Mandal, Lizhi Wang, James Lang, Alberto Passalacqua, Shankar Subramaniam, Guowen Song

**Affiliations:** 1Department of Apparel, Events, and Hospitality Management, Iowa State University, Ames, IA 50010, USA; ruili@iastate.edu (R.L.); mezhang@iastate.edu (M.Z.); yulinw@iastate.edu (Y.W.); liwenw@iastate.edu (L.W.); 2Department of Biological and Agricultural Engineering, University of California, Davis, CA 95616, USA; pxtang@ucdavis.edu (P.T.); gysun@ucdavis.edu (G.S.); 3Department of Design, Housing and Merchandising, Oklahoma State University, Stillwater, OK 74078, USA; sumit.mandal@okstate.edu; 4Department of Industrial and Manufacturing Systems Engineering, Iowa State University, Ames, IA 50010, USA; lzwang@iastate.edu; 5Department of Kinesiology, Iowa State University, Ames, IA 50010, USA; jlang1@iastate.edu; 6Department of Mechanical Engineering, Iowa State University, Ames, IA 50010, USA; albertop@iastate.edu (A.P.); shankar@iastate.edu (S.S.)

**Keywords:** respiratory protective device, filtration, healthcare worker, decontamination, fit, biocidal material

## Abstract

Infectious respiratory diseases such as the current COVID-19 have caused public health crises and interfered with social activity. Given the complexity of these novel infectious diseases, their dynamic nature, along with rapid changes in social and occupational environments, technology, and means of interpersonal interaction, respiratory protective devices (RPDs) play a crucial role in controlling infection, particularly for viruses like SARS-CoV-2 that have a high transmission rate, strong viability, multiple infection routes and mechanisms, and emerging new variants that could reduce the efficacy of existing vaccines. Evidence of asymptomatic and pre-symptomatic transmissions further highlights the importance of a universal adoption of RPDs. RPDs have substantially improved over the past 100 years due to advances in technology, materials, and medical knowledge. However, several issues still need to be addressed such as engineering performance, comfort, testing standards, compliance monitoring, and regulations, especially considering the recent emergence of pathogens with novel transmission characteristics. In this review, we summarize existing knowledge and understanding on respiratory infectious diseases and their protection, discuss the emerging issues that influence the resulting protective and comfort performance of the RPDs, and provide insights in the identified knowledge gaps and future directions with diverse perspectives.

## 1. Introduction

We have witnessed an increasing prevalence of infectious diseases, including SARS (2002–2003), H1N1 influenza (2009), and MERS (2012–2016) [1], that disrupt social activity and pose severe threats to human health and safety. Strikingly, as of 21 November 2021, the ongoing COVID-19 pandemic has resulted in more than 257 million infections and 5.1 million deaths (Figure 1) [2,3,4,5]. The human-to-human transmission of the above diseases occurs through both indirect (e.g., via fomites) and direct contact. The latter includes droplets (aerodynamic particle sizes >5–10 µm) and airborne transmission by droplet nuclei (i.e., virus-laden aerosols; aerodynamic particle sizes <5 µm) [6,7,8,9,10]. In addition to using several routes of transmission, SARS-CoV-2 has a higher infection rate and longer incubation period than other viruses [2,11,12,13,14]. Moreover, the COVID-19 pandemic continues to evolve rapidly. Recent reports have described the rapid spread of the SARS-CoV-2 variant Delta [15,16,17,18]; it has been found to exhibit much higher infectivity, 97–100% greater than that of the original epidemic strain [15,19,20,21], and is responsible for almost all (>90%) new infections around the world [15,16,17,18]. The emergence of new virus variants due to continued mutation poses challenges to infection control [22] and weakens the potential effectiveness of vaccines [23]. In fact, a crucial lesson learned by those high vaccination rate countries is that vaccine alone will not stop COVID [24].

The situation is exacerbated by pre-symptomatic and asymptomatic transmissions [25,26,27,28,29,30,31], i.e., individuals without symptoms can still be contagious and spread virus-laden droplets and aerosols through coughing, sneezing, speaking, or just breathing [32]. Indeed, research suggests that transmission of SARS-CoV-2 by pre-symptomatic or asymptomatic individuals is as high as 40–80% [27,33,34]. Even till recently, medical doctors still struggle with identifying pre-symptomatic or asymptomatic patients, because individuals usually will not undergo COVID testing until symptoms occur [35,36]. Data scientists also acknowledge that asymptomatic cases are a hidden challenge in predicting and modeling transmission patterns [37,38]. Bio-aerosols have been identified as an important route of airborne transmission by over 200 scientists around the world which highlights the importance of ventilation, physical distancing, and using RPDs as effective means for infection control [39]. Therefore, both the World Health Organization (WHO) [6] and US Centers for Disease Control and Prevention (CDC) [40] have recommended the universal wearing of face masks and coverings to achieve both source control and respiratory protection.

The high transmissibility and unique nature of COVID-19 emphasizes the vital role of respiratory protective devices (RPDs) as a component of personal protective equipment (PPE) in disease control. The adoption and use of different types of RPDs depends on the estimated exposure risk as well as on the specific occupational settings and scenarios. It must evolve along with the changing nature of the hazards and advances in technology and medical knowledge [41]. While firefighters and miners are traditional users of RPDs, at times of a pandemic workers in other occupations and even the general public must also adopt certain means of respiratory protection [6]. If respiratory protection is absent or used improperly, infections can spread rapidly, especially in constrained spaces with many coworkers and in the presence of airborne fluid suspensions, e.g., in the meat and poultry processing industry [42]. Today, filtering facepiece respirators (FFRs), medical/surgical masks, and various face coverings are the primary means of respiratory protection. According to the US National Academies of Sciences, Engineering, and Medicine, respirators, masks, and face coverings are all considered RPDs [43].

Another challenge during the COVID-19 pandemic concerns the RPD supply chain. While a supply shortage may lead to the inadequate protection of individuals and society as a whole, the sufficient supply, use, and disposal of RPDs may have a long-term environmental impact due to their low recyclability [44,45]. The latter can be remedied by decontaminating and reusing RPDs [46,47]. However, the SARS-CoV-2 virus can remain viable on material surfaces for a considerable period of time, including on textiles that are commonly used for medical purposes [48,49]. Thus, contaminated textiles and RPDs can be sources of secondary contamination through microbial transfer and re-aerosolization [50,51]. Although researchers have explored several decontamination methods to allow the reuse of RPDs, validated approaches and standards are needed to ensure their safety and efficacy.

The overall performance of RPDs is strongly associated with the type of hazardous environment, RPD design and engineering, and the different types of users and behaviors [32,52]. Therefore, research on respiratory protection is interdisciplinary in nature and requires a collaborative, convergent approach. Regardless of user group, RPD type, or usage practice, it is critically important to identify factors that directly or indirectly influence RPD performance, recognize emerging issues and knowledge gaps in respiratory protection, and gain insight into how to best address problems from multiple perspectives to be able to map future directions.

This review is aimed at researchers, manufacturers, end-users, regulatory agencies, and other stakeholders and intends to identify contemporary and emerging issues related to respiratory protection, specifically during a pandemic, while pointing out possible avenues for future research. We describe emerging and critical issues that directly or indirectly influence the performance of respirators, surgical/medical masks, and face coverings while highlighting knowledge gaps. Firstly, we discuss issues related to RPD protective performance and comfort, issues that are complicated by physical and physiological activities such as breathing resistance, facial pressure, high temperatures and humidity, as well as air exchange and CO_2_ over-inhalation. We also discuss user behavior and occupational factors affecting proper RPD usage. Secondly, we describe a numerical modeling approach to simulate the interaction between virus-laden media and RPDs that help to inform RPD design, development, and policy. Thirdly, we explore the decontamination and reuse of RPDs, stressing its environmental importance. Fourthly, we discuss future perspectives of novel materials including nanomaterials, biodegradable materials, and biocidal materials. Finally, we present an outlook of the next-generation RPDs for human respiratory protection.

## 2. RPD Performance, Issues, and Challenges

RPDs are designed to provide the wearer with a specific level of protection against a defined hazard [53]. Generally speaking, in the US, RPDs consist of various types of respirators that either require approval and certification by the National Institute for Occupational Safety and Health (NIOSH) [54] or medical/surgical masks that require clearance by the Food and Drug Administration (FDA) [55]. Face coverings such as homemade cloth masks, bandanas, and neck gaiters are not considered occupational PPE but fall into the RPD category during a pandemic [43,56].

Different types of RPDs provide different levels of preventive performance for different types of targeted hazards, such as dust and particulate matter (PM; e.g., asbestos or lead dust), fumes and smoke (e.g., metal fumes, soot), and mist or gas (e.g., chemical warfare gases, carbon monoxide) [53]. Atmosphere-supplying respirators supply clean air to the wearer when respirable air is absent or bio- or chemical hazards are present. In such cases, contaminated air is physically isolated from the human respiratory system [53]. By contrast, air-purifying respirators remove contaminants from ambient air by applying a filter. For example, FFRs are tightly fitting and utilize an integrated fibrous filtering piece to prevent inhalation of contaminants [53]. FFRs are designed to prevent transmission of PM and aerosols to a certain extent (including droplets if they are surgical FFRs, e.g., surgical N95). Typically, they are designed to be used once only, as the filtering piece is embedded in a molded structure and cannot be replaced. FFRs are the most popular type of RPD due to their low cost, accessibility, quick production, and simple management [1,57].

Surgical masks are also designed to protect the wearer from droplets and particles and to maintain a sterile environment in a clinical setting. However, because surgical masks do not tightly fit the wearer’s face, potential leakage around the mask perimeter can reduce its protective performance against small particles [53]. Thus, surgical masks alone do not provide effective inward protection against bioaerosols [53]. However, they can significantly reduce outward leakage of droplets which makes them popular for source control as it reduces the spread of viruses [58,59].

Filtering media used in FFRs and surgical masks are typically made from polymeric materials with a fibrous structure [60], such as nonwoven substances with fine (µm) fiber sizes, high packing density, and electrostatically charged [60]. Polymers that can be processed using nonwoven procedures (e.g., spunbonding, melt-blowing, electrospinning) and those with high electrical resistance and stability are the best filter choices [60]. They include polypropylene, polyethylene, polyacrylonitrile, polycarbonate, polystyrene, polyester, polyamides, and polyphenylene oxide [60].

The most common structure for surgical masks is a three-ply spunbond-meltblown-spunbond (SMS) structure. The center meltblown polypropylene layer serves as the principal filtering media [1,57,60]. In contrast, FFRs have an SMS structure with an additional supportive layer, usually made of polyester or modacrylic, to give the mask its three-dimensional shape for optimal fit to the user’s face. Due to the low moisture absorbency of polypropylene microfiber and its nonwoven structure, the outer layer of surgical masks/FFRs is usually water-resistant and repels body fluids and droplets. The inner layer, with some treatment, can absorb some of the moisture released by the wearer to maintain comfort [1,57,60].

Unlike surgical masks or FFRs, face coverings used by the general public are made from various materials, designs, and structures. Currently, there are no specifications for face covering design or performance. Since the onset of the COVID-19 pandemic, efforts have gone into compiling guidelines on mask design and material selection for the general public that also include performance evaluation methods and requirements for manufacturers. For example, the American Association of Textile Chemists and Colorists (AATCC) developed a general guidance document for textile face coverings [61], and ASTM International is developing a standard specification for face coverings considering protection and comfort levels [62]. However, comfort level is loosely regulated by this ASTM standard, as only inhalation resistance, but no thermal comfort or moisture management properties are considered.

Evidence of pre-symptomatic and asymptomatic transmission of the SARS-CoV-2 virus [26,27,63,64,65] suggests that wearing face coverings can substantially reduce viral spread in a community. A recent simulation demonstrated that the spatial virus spread in an open, windy environment can considerably exceed the physical distancing requirements (currently 2 m) [66,67,68]. The use of face masks and coverings is encouraged and has been demonstrated to be effective for controlling disease transmission [6,40,69,70,71].

However, the effectiveness of RPDs can be affected by various factors. Filtration efficiency depends on the nature of the hazard, the environmental conditions, as well as the type of filter material and structure. Its protective performance also depends on the facepiece fit and seal while various user and usage factors affect comfort (Figure 2).

### 2.1. Filtration Mechanisms and Efficiency

Filtration efficiency provided by RPDs is dynamic in nature and influenced by the inherent properties of the filtering material, external hazard factors such as bioaerosol size and airflow, and human physiology (Figure 2). The ability of a nonwoven filter to block and trap particles primarily depends on a range of mechanical and electrostatic mechanisms, namely sedimentation by gravity, inertial impaction, interception, diffusion, and electrostatic attraction and induction [1,72,73,74,75,76] (Figure 3).

Gravitational sedimentation occurs during the early stages of expelling droplets, when large aqueous particles (>1–10 µm) sediment due to the gravitational force [77]. Inertial impaction is the principal mechanism to trap aerosol nuclei (~1–5 µm) whose inertia is too high to follow the airflow (which moves around the fiber) and causes them to collide with and adhere to the fine filter fibers [78]. Smaller particles (0.1–1 µm) moving with the airflow and can only be intercepted within a range of one particle width of a fiber surface. Even smaller (nanoscale) particles move randomly (Brownian motion) and can bounce into and be trapped by the filter media [78], which constitutes the filtration by diffusion mechanism. These mechanical filtration processes are least effective for particle sizes of ~0.3 µm [74,79]. This size is therefore termed the most penetrating particle size (MPPS) which is the typical particle size in challenge agents used in filtration efficiency evaluations [54,80]. Electrostatic attraction and induction are additional filtration mechanisms that complement mechanical filtration. Electret filter materials are capable of absorbing oppositely charged particles onto the filter fiber while non-charged particles are attracted by induced dipoles that can remove them from the airflow [77,79,81,82]. While these mechanisms act on a wide range of particle sizes, they tend to be more effective for the low airflow velocities that occur during respiration when a person is at rest [77].

Viruses are usually in the size range of 20–400 nm, bacteria around 0.2–2 µm, and PM in the micron range (PM_2.5_–PM_10_) [57]. While SARS-CoV-2 is a spherical virus with sizes from 60–140 nm [83], they are usually embedded in droplets or attached to aerosols or other small particles (e.g., pollutants), which means that in order to prevent its spread, RPDs need to be able to capture and filter aqueous and solid particles in the submicron to micron size range.

Environmental factors also affect RPD filtration efficiency. For instance, airflow speed or face velocity against the respirator play key roles. In general, higher airflow rates reduce filtration efficiency [77,84,85]. Higher airflow speeds increase the inertia of large particles that are then captured by impaction [79], whereas lower airflow increases the loading or residence time of particles which are then captured by diffusive or electrostatic mechanisms [79,86]. Because interception and diffusion are the most dominant filtering mechanisms in the submicron range [72], lower airflow will result in higher filtration efficiency. In contrast, variable airflow velocities caused by varying respiration volumes and frequencies will reduce filtration efficiency [79,86,87,88], in other words, the higher the breathing frequency, the lower the filtration efficiency [79].

Environmental conditions such as humidity also influence RPD filtration efficiency but are less studied [84]. Increased humidity may reduce the charges of the electret filter and particles, thereby reducing filtration efficiency by electrostatic attraction and induction [89]. Increased humidity may also lead to higher capillary forces allowing aqueous particles to penetrate filter fibers (wet penetration of viruses) [90,91]. Recent studies demonstrated that MPPS and filtration efficiency depend on the ambient relative humidity [92].

An understanding of these factors is required to be able to estimate RPD filtration efficiency in different usage environments and scenarios. E.g., when a person is operating under a high workload, salient changes in their breathing frequency and pattern can reduce filtration efficiency. The following table summarizes environmental and human factors that influence RPD filtration efficiency (Table 1).

### 2.2. Filtration Efficiency Requirements and Testing Methods

Regulatory or guidance standards, product specification/requirement standards, and test method standards for evaluating filtration efficiency and other key properties of FFRs and surgical/medical masks vary across organizations and countries. In the US, the Occupational Safety and Health Administration (OSHA), NIOSH, and FDA issue regulatory, guidance, and specification standards, whereas the American National Standards Institute (ANSI), NIOSH, and ASTM International issue test method standards for PPE. In the European Union (EU), the European Commission regulates PPE, and the European Committee for Standardization adopts and issues requirements and test method standards (Table 2 and Table 3).

As can be seen, the requirements and test methods for different types of RPD vary considerably. A particular emphasis lies on filtration efficiency which is evaluated using various challenge agents at varying intensities, concentrations, and flow rates. As a result, some standards are much stricter than others [110]. For example, the NIOSH aerosol test, which was designed to evaluate FFRs in an industry setting, utilizes much finer aerosols and at higher concentrations and flow rates than the ASTM standard. Hence, a serious problem arises due to the mismatch between performance needs and evaluation methods. That is, current FFR test methods set a very conservative bar and do not represent the environment and usage scenarios for occupations such as healthcare workers.

While the surgical N95 FFR used by US healthcare workers provide superior particle filtration, their wearing comfort could be improved as evidenced by the sometimes low user compliance [52]. Indeed, much industry debate currently revolves around whether sub-N95 filtration masks, which are more comfortable and pose less of a hindrance to certain work activities, can provide an adequate level of protection for healthcare workers. So far, however, no FFRs have been proven to provide an adequate balance between protection and wearer comfort in hospital environments with the wearer performing healthcare-specific work activities.

Additionally, requirements and evaluation methods for RPDs used in similar settings (e.g., N95 and FFP2 FFRs) vary across countries. For example, compared to the US, the EU utilizes a lower particle concentration and higher flow rate to assess FFR filtration efficiency but has more detailed breathing resistance and carbon dioxide (CO_2_) content requirements. By contrast, the US has stricter requirements for surgical/medical masks than the EU that only requires Type IIR masks for fluid splash resistance. Thus, different standards and usage recommendations for FFRs may cause confusion, reluctance in adoption, and uncertainty in protection. Although the FDA issued emergency use authorizations (EUA) for FFRs manufactured according to the standards from other countries, one NIOSH report revealed that some of these FFRs barely pass US requirements and sometimes are untestable due to their completely different construction [111].

Information to guide the general public’s choice of fabric, design, and construction of face coverings is of critical importance. Unfortunately, there has been no standard test method to evaluate the various fabric combinations. With the ongoing COVID-19 pandemic and in order to save FFRs and surgical masks for healthcare workers, adequate materials for making face coverings for the public need to be identified as quickly as possible. Several studies attempted to evaluate the filtration efficiency of the various fabrics and fabric combinations, but their results so far have been largely inconsistent (Table 4).

Although these studies have successfully evaluated the filtration efficiency of common fabrics and cloth masks and compared them with surgical/medical masks and N95 respirators, much is still unknown. In particular, the specific test conditions and challenge agents varied considerably between studies, resulting in low replicability and a lack of generalizability. Additionally, no existing evaluation method can replicate the specific exposure conditions experienced by the general public. Although some specific conditions have been simulated or modeled (e.g., open spaces, inside a flight cabin, inside a ventilated room [66,67,112,113,114,115]), real-world exposure conditions are diverse and have not been systematically investigated. Thus, the evaluation conditions used in these studies often fail to represent the real world.

Moreover, current evaluation methods focus on assessing the filtration efficiency of textile materials only, whereas other key performance measures such as fit, breathability, comfort, wearability, washability, and the potential reduction in filtration efficiency during use and after multiple washes are critical determinants of cloth mask efficacy. Thus, product requirements and test methods targeting the entire usage lifespan should be developed and updated with actual usage scenarios for the general public.

Furthermore, a key route of virus transmission—droplet splash—has not been simulated, despite being identified as a transmission route of SARS-CoV-2 and a mechanism that produces fomites. The ability to repel fluids (e.g., droplets generated by coughing and sneezing) and filter out submicron particles are both important for source control and provide personal protection on a daily basis [6,40]. A wet face covering, or one that becomes moist during usage, can lead to a high risk of infection due to wet penetration of the virus [90,91]. Fortunately, this issue is being investigated through experiments, simulations, and modeling studies.

Researchers employ human trials, experimental simulations, computational fluid dynamics (CFD) simulations, and numerical modeling to understand, describe, and predict the trajectory, dispersion, evaporation, and sedimentation of large droplets and small aerosols generated by coughing and sneezing. Some studies use a high-speed camera system to capture coughing and sneezing behaviors and characterize airflow, fluid, and particle dynamics which are then modelled to investigate aerosol transmission in various environmental conditions and identify potential infection or contamination scenarios [116,117,118,119,120,121].

**Table 4 polymers-13-04165-t004:** Summary of filtration efficiency evaluations of the various materials used to fabricate homemade mask/face coverings.

Tested Items	Test Condition	Results	Conclusions
N95 filter media, cloth masks, sweatshirts, T-shirts, towels, scarfs	**Method**: material testing **Agent**: NaCl aerosol **Size**: polydisperse particle median diameter 75 nm, 10 levels of monodisperse particles diameter 20–400 nm **Flowrate**: 33 L/min and 99 L/min **Concentration**: NA	In both flowrate conditions, N95 filter media had less than 4% penetration, while other tested items had 40–90% penetration for polydisperse particles and 9–98% penetration for various sizes of monodisperse particles	Common fabric material only provides marginal protection against small particles, filtration efficiency for different particle sizes varies significantly [95]
Surgical mask, T-shirt, scarf, tea towel, pillowcase, vacuum cleaner bag, cotton mix, linen, silk	**Method**: material testing **Agent**: bacterial aerosol (*Batrophaeus*), viral aerosol (Bacteriophage MS2) **Size**: 0.95–1.25 µm, 23 nm **Flowrate**: 30 L/min **Concentration**: 10^7^ colony-forming units, 10^9^ plaque-forming units	For bacterial aerosol, filtration efficiency ranged from 58% to 96%; for viral aerosol, filtration efficiency ranged from 51% to 90%	Surgical mask, double layer tea towel, and vacuum cleaner bag had similar filtration efficiencies (>94% for bacteria; >85% for viruses); double layer T-shirt did not offer any improvement over single layer [122]
N95 masks, surgical masks, cloth masks	**Method**: product testing with head manikin **Agent**: polystyrene latex, diluted diesel combustion particles **Size**: 5 levels of monodisperse particles 30 nm to 2.5 µm, diesel particle size < 500 nm **Flowrate**: 8 L/min and 19 L/min **Concentration**: 2.84 × 10^3^ to 2.77 × 10^5^ no./cm^3^; 4.13 × 10^2^ to 2.66 × 10^4^ no./cm^3^	Cloth mask filtration efficiencies ranged from 15–57% for diesel particles and 39% to 65% for latex particles; disposable surgical masks had efficiencies of 78–94% for latex and 79% for diesel particles.	N95 masks were effective at removing most test particles; surgical masks were surprisingly effective for all test particles; cloth masks only had a marginal filtration efficiency [123]
N95 masks, surgical masks, various fabrics (cotton quilt, cotton 80 TPI, cotton 600 TPI, flannel, chiffon, natural silk, synthetic silk, satin, spandex, polyester)	**Method**: material testing **Agent**: NaCl aerosol **Size**: polydisperse sizes ranging from 10 nm to 6 µm **Flowrate**: 35 L/min and 90 L/min **Concentration**: NA	At the lower flow rate, several fabrics achieved the same filtration efficiency as N95 and surgical masks (75–99%); at the higher flow rate, N95 maintains a high efficiency (>94%) while the other materials exhibit a significantly reduced efficiency (14–64%), especially for particles <300 nm	At the lower flow rate, fabric combinations such as cotton-silk, cotton-chiffon, and cotton-flannel had filtration efficiencies above 80% irrespective of the particle size; the number of fabric layers and the fabric density (i.e., threads per inch) both affected filtration efficiency [77]

In other research, droplet and aerosol generation is simulated using devices such as spray guns to study the transport of aqueous particles [124,125]. As these types of simulations are easier to replicate and repeat in different environments, they can be used to further explore the influence of environmental conditions on particle transmission. Based on the data generated by such approaches and human subject testing, many researchers have been able to validate mathematical models that can parameterize the different geometries of the buccal/nasal passage, physical properties of human sputum, environmental conditions, and viral concentrations in the upper respiratory tract [126,127,128,129].

Advances in evaluating the performance of RPDs began with measurements of the size distributions of actual droplets and aerosols generated during respiration, speaking, coughing, and sneezing. Research has shown that individual human-generated droplets typically come in sizes that range from a few microns (2–7 µm) to several hundred microns (200–900 µm), whereas human-generated aerosols tend to be in the submicron range (0.3–1 µm) [130,131,132]. In order to be effective, respiratory protection should therefore be able to repel micron- to millimeter-sized fluid splashes and stop the penetration of submicron aerosol suspensions.

Researchers also developed lab-based breathing, coughing, and sneezing simulators to evaluate the protective performance of face shields, respirators, and face coverings in different environmental conditions [133,134,135,136,137]. A recent study by Lindsley et al. using cough simulator indicated that N95 respirators provide superior protection against aerosols (0.6–7 µm, >99%) compared to medical and cloth masks (51–60%) [138]. However, face shields provide no meaningful protection (2%) if used alone, suggesting that face shields should always be used in combination with FFRs or surgical/medical masks [139].

Some researchers used cough simulators to evaluate the effectiveness of air ventilation systems in confined spaces [140]. Others investigated the influence of thermal buoyancy (resulting from human body temperature) on particle movements using a breathing simulator and thermal manikin [141,142,143]. National agencies including the US National Personal Protective Technology Laboratory (NPPTL) have used head manikins to assess respirator fit [144,145,146]. Furthermore, the influence of human head movement on the performance and face seal of face coverings has been studied recently using a moving head form [147].

These advances provide a glimpse of a future where the protective performance of RPDs and face coverings can be evaluated in a more comprehensive manner that considers the nature and intensity of hazards, environmental conditions, human physiological features, and activities including specific occupational tasks. However, at present, no systematic evaluation tool or associated regulation or test standard exists. Such tools and standards, if properly implemented with adequate risk assessment, would aid the development of suitable RPDs that offer sufficient protection and address the need for improved comfort and ergonomics for specific occupations.

### 2.3. Fit Requirements and Test Methods

A well-fitting FFR is critical for healthcare workers during the COVID-19 pandemic, especially those who perform aerosol-generating procedures (AGPs). In the US, FFRs intended for occupational use must pass a fit capability test to assure that their design provides a good fit for users [148]. In addition, each user must undergo an individual fit test to ensure that the FFR achieves a proper seal—both at the time of initial employment and annually thereafter [52,53,97].

ASTM F3407, the standard test method for respirator fit capability for negative-pressure half-facepiece particulate respirators, employs a quantitative fit test protocol in accordance with NIOSH face panel data. This test protocol is based on OSHA regulations for respirator fit testing [97]. Essentially, fit capability determines the percentage of the population that can achieve an adequate seal for a given respirator model. In the US, an N95 respirator must provide an adequate fit for at least 95% of human subjects in a test panel [149,150].

Airborne protection can only be guaranteed if an FFR fits tightly to the wearer’s face without leakage [151]. Protection against airborne pathogens is reduced if a leak is present, as the wearer will breathe some unfiltered air. Hence, an adequate fit is pivotal for airborne protection regardless of the FFR’s filtration capacity [152]. Due to their two-dimensional structure and loose fit, surgical/medical masks provide much lower protection against aerosols even though their filtering piece is made of the same polypropylene meltblown material as in FFRs [152,153]. As facial contours vary, the design, size, and shape of an FFR are key to achieve a good fit in a large percentage of the population [154]. Therefore, an RPD fit test is mandatory for individuals who need respiratory protection in their job and is regulated by OSHA [155].

OSHA follows a specific protocol for fit testing that includes both qualitative and quantitative assessments [155]. The qualitative fit test challenges an FFR wearer with bitter- or sweet-tasting aerosolized agents (e.g., denatonium benzoate, saccharine) in a test hood. The wearer’s inability to taste these agents is indicative of a good fit of the FFR without leakage [156]. A weakness of this method, however, is that individuals have different sensitivities to these challenge agents. As a result, this relatively low-cost method is gradually being replaced with a more objective and quantitative test method (also required by OSHA) [155].

This quantitative fit test uses an environment containing a specific concentration of a non-harmful aerosol (NaCl) and the goodness of fit of the FRR is calculated based on the ratio between the substance concentrations inside and outside the mask. NIOSH considers a value of >100 indicative of a sufficiently good fit (i.e., no leak) [154,157]. As the detector cannot differentiate between particle penetration via facial leakage or through the filter itself, FFRs often wrongly fail the fit test. To address this problem, TSI Inc. developed the PORTACOUNT tester, which incorporates an N95-companion technology that ignores very small particles in its particle count and only focuses on relatively large particles that cannot penetrate the FFR filter but whose presence can only be the result of FFR leakage. This provides a much more accurate determination of the fit factor.

The fit performance of an FFR depends not only on how well its shape and size match the wearer’s face contour but also on body movements and facial expression changes. Therefore, OSHA fit test protocols include specified movements, such as turning one’s head up and down and from side to side, reading a paragraph of text, and bending over at the waist [155].

In addition to a proper fit test, individuals are required to self-perform a leakage check every time they put on an FFR to ensure there is no detectable leakage under positive or negative pressure [156]. However, this self-check cannot replace a mandatory fit test. Indeed, NIOSH showed that the protection of N95/FFP2 masks improved from 67% without fit testing to 96% with fit testing, which clearly demonstrates the necessity of proper fit testing [158].

#### 2.3.1. Fit and Inward Leakage Issues

Although fit testing and self-checks are strictly regulated and required, some issues related to fit and potential leakage still exist. The first problem is that traditional FFR designs do not fit equally across different ethnicities and gender groups, resulting in high rates of fit test failure for certain populations. Asians and females typically experience the poorest fit, as they are underrepresented in the fit test panel [150,151,156,159,160]. One study reports that on average only 60% of Asian females pass the fit test [161].

In addition, passing a fit test with a given respirator does not guarantee dynamic fit while performing occupational tasks [159,162]. Even when leakage is properly self-checked every time an FFR is donned, changes in its fit can occur during a work shift, especially during the very extended shifts put in by healthcare workers in a pandemic. The standardized movements used in the quantitative fit test do not fully represent the movements performed during specific occupational tasks, i.e., that the goodness of fit during work scenarios could be overestimated in these tests. Recent studies have demonstrated various protection failures of fit-tested RPDs for different medical activities, especially for repeated, high intensity activities such as cardiopulmonary resuscitation (CPR) [163,164,165,166]. Protection was found to be reduced by 30% because of a compromised face seal [163,164]. Furthermore, previous research stressed that FFRs can interfere not only with patient-doctor communication but also with the performance of medical/clinical treatments due to vision hindrance [52].

Another issue stems from the low (annual) frequency of the fit test, as face sizes may change during a year which could lead to poorer fit. However, more frequent fit tests would increase operational costs and could cause difficulties for hospital PPE management, especially at times of PPE shortages, e.g., during a pandemic. The adoption of new respirator models also requires additional fit tests for all employees, which presents an additional strain to hospital administrators [149,167].

These issues demand updated fit test protocols and equipment that can be easily utilized in specific occupational fields. New FFR face-seal designs that can fit a broader ethnic range of individuals should be explored; for example, some studies achieved an improved fit by adding a new structure inside the FFR perimeter [168]. Moreover, real-time fit sensing and leak warning technology incorporated into the FFRs also seems promising [169,170,171]. Additionally, advanced materials should be incorporated into FFR construction to achieve a custom or adaptive fit for wearers. Ideally, customized FFRs that can adapt to various facial movements and expressions should be developed. In a pandemic, it would also be desirable that FFRs could be disinfected easily and have replaceable filters to extend their reusability.

For the general public using medical/surgical masks or face coverings, a perfect fit cannot be realized. However, adding simple means to partially eliminate openings along the perimeter of cloth masks can significantly improve filtration efficiency [172]. Thus, guidance should be provided on the best design and construction of masks that fit most people with minimal openings.

#### 2.3.2. Fit and Inward Leakage Issues

A properly fit-tested FFR should provide satisfactory protection against both inward and outward transmission of droplets and aerosols if the headband provides enough force to assure a tight fit regardless of head or face movement. However, surgical/medical masks and face coverings do not fit tightly and thus cannot provide a high level of protection against inward and outward particle transmission [173,174]. As such, potential outward leakage is a critical concern because it could spread virus-laden aerosols and contaminate the environment and objects. To date, only few studies attempted to evaluate the effectiveness of surgical/medical masks and face coverings in preventing droplet or aerosol spread. For example, using an automated manikin head and aerosolized endospores or vegetative cells to evaluate outward leakage, Green et al. (2012) found that 50–76% of aerosols were arrested by surgical masks [175]. Also, a recent study with human subjects by Leung et al. (2020) demonstrated the effectiveness of surgical masks in reducing the emission of virus particles through exhalation [176], corroborating another study that had concluded that wearing surgical masks reduced virus aerosol shedding [177]. Thus, although homemade masks lack the fit and blocking abilities of surgical masks, they can still substantially reduce the expulsion of microorganisms [122].

However, a recent study investigating the efficacy of masks and face coverings in controlling outward aerosol emission found that in certain cases wearing homemade masks can amplify the emission of aerosols [178]. These aerosol particles may not originate directly from respiration but from the face covering material [178]. As this study is unique in its finding that contamination was increased by wearing homemade masks, more research is needed to further assess this claim. Nevertheless, this study highlights several key issues that must be addressed. Firstly, it suggests that outward leakage can be amplified if the wearer engages in strong expiratory activities, such as coughing and sneezing, which substantially increases perimeter leakage and air flow. Secondly, it suggests that without proper pretreatment, friable fibers and other particles in the face covering material can be shed and spread. Hence, homemade face coverings should avoid single layer low density fabrics (e.g., Jersey knit fabric) and adopt multilayer, high density fabrics (e.g., twill/ripstop woven fabric) instead.

These findings underline the complexity of relying on universal mask wearing as a key measure for infectious respiratory disease control, as mask material, design, and wearer activity and compliance all impact the effectiveness of this strategy. This further emphasizes the importance of developing a proper tool for evaluating the outward leakage of RPDs, especially medical/surgical masks and face coverings that are more important for source control. Specifically, establishing a list of requirements regarding fabric type, structure, and layers based on scientific and technical data for homemade face coverings is of great importance. Without such a tool, associated research, technical data, and requirements, effective standards for the design and material selection cannot be realized.

### 2.4. Discomfort Related to Breathing Resistance and Air Exchange

An important parameter affecting comfort and elevated CO_2_ contraction through rebreathing is RPD airflow resistance [179]. Airflow resistance is characterized by differential pressure, which is an indicator of the physical properties of the filter material or layers in the RPD. The airflow resistance of the filter depends on the type of textile-based porous media and the surface area of the microfiber. Increased airflow resistance of respirators has resulted in respiratory fatigue and impaired work efficiency.

Whereas flow resistance can be measured with a standard setup from the pressure drop across the facemask; breathing resistance can be evaluated using both objective and subjective methods [180,181,182]. Pioneering work on breathing resistance measurement by Lee and Wang [183] assessed nasal airflow resistance during breathing cycles with standard rhinomanometry and nasal spirometry. For N95 respirators, they found average increases in inspiratory and expiratory flow resistance of 126% and 122%, respectively. Subsequent studies comparing N95 respirators and surgical masks [184] further demonstrated that N95 respirators induce higher post-wearing nasal resistance recovery routines. However, wearing duration was only 3 h, which does not simulate the conditions during a pandemic.

Yao et al. investigated twelve types of RPDs with different characteristics using a dynamic manikin head system to simulate the breathing process [185]. Compared to a folding mask, face masks with a respirator valve and cup-type masks both exhibited a less variable breathing resistance. Also, cotton masks were showed a lower variability in breathing resistance than nonwoven fabric masks. Another study performed by NPPTL also demonstrated increased breathing resistance because of exhaled moisture build up during long term respirator wearing [186]. However, further research is needed to assess human sensations and possible health effects of this breathing resistance.

The primary function of the human respiratory system is to ensure efficient gas exchange with the environment, providing oxygen (O_2_) to cells and removing CO_2_, the major byproduct of cellular metabolism, from the body [187]. This function is achieved by the movement of ambient air in and out of the lungs (i.e., ventilation) and gas exchange through the semipermeable lung membrane (i.e., external respiration) [188]. Normally, humid, hot air that is rich in CO_2_ is exhaled through the mouth and nostrils during expiration, and air from the immediate surrounding is inhaled during inspiration. When wearing a respirator or face mask, however, a dead space between the mouth/nostrils and respirator/face mask is created, which drastically alters the pattern of air flow during respiration [179]. As a result, the wearer may experience discomfort and inhale air with elevated CO_2_ concentrations, the degree of which depends on the type of RPD, duration of RPD use, environmental conditions, and type of human activity [179].

Rebreathing air with elevated CO_2_ is a major concern with RPD use, particularly when used for prolonged periods as occurs during a pandemic. Elevated CO_2_ inhalation can affect many physiological systems and cause discomfort, fatigue, dizziness, headache, muscular weakness, and drowsiness [179]. These symptoms can further intensify as work activity (i.e., metabolic rate) increases [179]. The flow resistance and dead space produced by RPDs affects inspiratory and expiratory resistance and are strongly related to breathing patterns and CO_2_ retention [189].

During a pandemic, respirator reuse is an important issue to consider. To reuse an N95 respirator, the Institute of Medicine recommends that healthcare workers use a surgical mask over their N95 respirator to avoid its surface contamination. A study applying the NIOSH Automated Breathing and Metabolic Simulator examined how six different levels of work intensity affected breathing resistance and the amount of inhaled CO_2_. Under most work rates, the respirator-surgical mask combination affected inhaled breathing quality with lower levels of energy expenditure and breathing resistance with higher levels of energy expenditure. The respective averages from all 30 NIOSH models of inhaled CO_2_ and O_2_ were 2.7% and 17.1% without surgical mask and 3.0% and 16.7% with surgical mask. The presence of a surgical mask thus increased the inhaled concentration of CO_2_ but did not change peak exhalation or inhalation pressure [181,190]. It should be noted, however, that the different approaches used by NIOSH and the EU to determine respirator average CO_2_ concentrations yield different results for the same respirators [191].

In another study, Smith et al. investigated the effect of speech and work rate on rebreathing by monitoring the amounts of expired and inspired CO_2_ in workers, who had been trained in the use of RPDs, while performing a graded exercise test. Speech and low work rates were associated with higher levels of CO_2_ rebreathing [192]. Similar studies demonstrated that increased levels of CO_2_ are not completely removed from the rebreathing space of RPDs [193]. These findings call for more research to further understand how increased CO_2_ rebreathing is related to RPD material and design and how it affects human physiology and health.

### 2.5. Thermal, Moisture, and Physical Discomfort

RPDs should provide an adequate transfer of metabolic heat, exhalation, and sweat vapor to the ambient environment. However, current RPDs are not subject to any thermal or moisture management standards, which often results in poor wearer comfort. In general, flexible textile fibers and yarn-based fabric materials provide a better thermo-physiological comfort. Face coverings made from natural fibers (e.g., cotton, linen, silk) may provide better comfort because of their thermal properties and moisture absorbency. In contrast, RPDs made from synthetic fibers (e.g., polypropylene, polyester, polyacrylonitrile) are usually associated with higher thermal resistance and lower moisture absorbency. As a result, an effective transmission of metabolic heat and generated moisture is difficult to achieve with synthetic fibers. Thus, RPD fabrics that trap moisture not only reduce wearer comfort [194] but also influence filtering efficiency [92]. In extreme conditions, moisture condensation may occur, which can cause facial irritation, wetness, virus wet penetration, and noncompliant behavior (e.g., adjusting the RPD during usage) [91,195,196].

A number of studies have theoretically and empirically demonstrated that wearing tightly fitting RPDs (e.g., N95) significantly increases facial temperature and humidity, which can quickly lead to fatigue and thermal stress [180,197,198,199,200,201,202], whereas loose fit surgical masks are typically associated with less severe physiological impacts [180,203]. While wearing RPDs for a short term under low to medium workloads may not have any direct negative impacts on human health [204], prolonged RPD use has been associated with a reduction of both heat and moisture loss and increased breathing resistance which may lead to adverse psychophysiological, thermoregulatory, and cardiovascular health effects [205,206,207]. It is thus essential to use materials with a higher thermal conductivity and vapor permeability to provide better thermal and moisture management abilities [208].

While exhalation valves can significantly reduce RPD microclimate temperature and humidity, they are not suitable for a clinical setting [198,209,210]. Therefore, new RPD designs are required that provide increased comfort, lower hindrance of work activities, and adequate protection in clinical settings [52,211]. Several researchers have attempted to answer this call by designing FFRs with a form of air-conditioning or air-cycling to improve wearer comfort. However, these efforts are still at an exploratory stage and have led to FFRs that are still heavier and more costly [212,213].

Another issue is the physical discomfort caused by the applied facial pressure when wearing an RPD. On the one hand, the pressure should be sufficiently high to guarantee a tight and stable respirator seal. On the other hand, too much pressure can lead to facial irritation and pain, thus reducing compliance. Strap tension, placement, orientation, friction, and face seal material softness have all been shown to significantly affect contact pressure [214,215]. Additionally, head movements may amplify contact pressure in certain regions of face [216,217]. The tight-fitting straps and head harness can elicit symptoms such as acne [218], nasal bridge scarring [219], facial itching, lightheadedness, and headaches [220,221,222]. In one study, up to about 80% of healthcare workers experienced headaches while wearing an N95 respirator [220]. In a different study, over one-third of workers reported more than six headaches in one month [221]. Although confounding factors (e.g., increased stress, longer working hours, and shift work) may contribute to the increased frequency, headaches were more likely to develop after having worn a mask for >4 h and in individuals that had pre-existing headaches [221]. The level of protection afforded by the mask is a high priority; however, it is also important to consider comfort, especially with long-term mask use as encountered during a pandemic. These findings call for improved RPD designs to achieve a better dynamic fit and wearer comfort. It is critical to balance the protection and wearing comfort so that extended usage of RPD can be achieved without interfering with human functions, especially during a pandemic.

## 3. User Groups, Specifications, and Key Issues

### 3.1. Healthcare Workers

For healthcare workers, it is often impossible to implement the typical measures to reduce exposure (e.g., elimination or substitution of hazards, isolation, shortened working hours) [52]. As a result, RPDs such as surgical N95 respirators become the last, albeit least desirable, line of defense for healthcare workers as they strongly rely on user compliance and many other factors to assure their effectiveness [52,72]. Compared to the general public, healthcare workers have been disproportionately infected during recent infectious disease outbreaks [223,224,225,226]. As of 21 November 2021, the COVID-19 pandemic had resulted in over 2638 deaths and 745,311 confirmed infections among healthcare workers in the US alone [227]. In particular, nurses and emergency medical technicians (EMTs) working in emergency departments have higher infection rates than those working in an intensive care unit (ICU) or other inpatient settings [228]. Additionally, ~50% of healthcare workers who had tested positive for COVID-19 were asymptomatic, stressing the importance of compliant and consistent usage of RPDs [27]. However, wearing RPDs may impose time constraints and discomfort, which can interfere with healthcare activities and communication. Furthermore, patient care involves many AGPs which further increases the risk of infection [229,230].

Potentially hazardous AGPs include tracheal intubation, non-invasive ventilation, CPR, nebulization of drugs, and ventilator disconnections [231,232,233]. These procedures are more likely to be conducted on patients with severe symptoms who consequently have much higher viral loads in their respiratory tracts. The prevalence of infection is usually higher among healthcare workers involved in AGPs [229]. As a result, healthcare workers performing AGPs are always required to wear full PPE as a precaution against airborne viruses [6,234,235,236,237].

Because of their highly hazardous work environment and risky occupational tasks, frontline healthcare workers have an elevated COVID-19-positive rate [238]. Furthermore, healthcare workers who reported inadequate PPE or PPE reuse were at a higher risk [238]. Evidence has shown that the pressure to care for patients in high-risk settings without adequate access to PPE partially explains why healthcare workers from minority ethnic groups were disproportionately affected by COVID-19 [238]. These same observations have also been reported for the SARS epidemic [239,240]. Healthcare workers involved in AGPs were 6.6 times more likely to be infected [230]. PPE shortages, reuse, and extended use during a pandemic can further exacerbate the risk of infection [241,242].

During the SARS outbreak, transmissions occurred despite the usage of PPE suggesting an incorrect or inconsistent use of PPE and non-fitted FFRs [158,239,240,243]. These cases were not limited to individual countries which suggests that this failure of PPE and FFRs may be universal to any hazardous environment, but especially to healthcare settings.

In addition to the high infection and mortality rates [244], healthcare workers exhibited high levels of psychological distress, such as self-reported anxiety, depression, and symptoms of post-traumatic stress [228]. These burdens appear to be even more severe in developing countries, where medical resources are often scarce [245]. Thus, especially during a pandemic, it is critical to improve the protection of healthcare workers to be able to maintain a healthy workforce that can attend to the high patient numbers.

Thus, we need to develop a specific type of RPD that is specifically designed for healthcare settings and the particular hazards and occupational activities (including AGPs) encountered there. At present, although existing guidelines still recommend the use of N95/FFP2 FFRs when performing AGPs, devices that afford improved protection and comfort, such as powered air purifying respirators (PAPRs), are already being used in the US.

### 3.2. Atypical RPD-Users and the General Public

For professions without traditional RPD use and the general public, it is critical to evaluate the specific exposure risk before recommending RPD adoption and usage guidelines. Special attention should be paid to people who work in constrained spaces in close proximity to many coworkers or customers, e.g., in the meat and poultry, transportation and warehouse, and retail and service industries. Failure to employ proper protection may result in concentrated infections among a large number of workers [42,246,247].

Compared with healthcare workers, the general public faces lower risks of infection if proper physical distancing and use of face coverings are practiced. However, exposure control measures are hard to implement among the general public due to various political, economic, social, and cultural reasons. As a result, the adoption and use of different types of respiratory protection are usually more focused on source control rather than perfect personal protection.

On one hand, medical/surgical masks and face coverings cannot provide the same level of protection as fit-tested FFRs due to their material, design, and loose fit. On the other hand, a very small proportion of infected people (so-called super-spreaders) can cause vast spreading of the virus, highlighting the importance of source control [248,249]. For example, studies estimate that 10% of infected people are responsible for 80% of disease spread [250] or that 8% of infected people are responsible for 60% of secondary cases [251]. In the US, ~2% of infected people are directly responsible for 20% of all infections [249]. As a result, suitable education and guidance should be provided to the general public to achieve compliance, which is key to effective source control.

A critical issue for universal mask-wearing is the lack of guidance for the general public to make and purchase face coverings that are protective. Not all face coverings are protective, and some may even concentrate particles or contaminates on the surface of the covering materials [123,178]. Although many studies have attempted to evaluate the filtration efficiency of various fabrics and their combinations, the results are usually inconsistent (Table 4).

Additionally, these test results are difficult to generalize due to the particularities of testing conditions and materials used. Hence, an important need exists to create standards and guidance for proper material selection, mask design and construction, and suitable evaluation criteria to ensure proper protection and source control for the general public.

### 3.3. Protection of Vulnerable Populations: Children and the Elderly

Children are a vulnerable segment of any population [252], and the number of children with COVID-19 infection has increased dramatically even though most of their symptoms are mild [253,254]. Children differ substantially from adults in their biological/physiological responses to exposures of environmental hazards. Therefore, simply considering children as “small adults” is not appropriate; rather, a systematic and comparative approach to COVID-19 risk assessment is required [255,256,257]. Compared with adults, children tend to be more physically active and spend more time engaged in indoor and outdoor activities (i.e., higher respiratory rates). Hence, due to the higher minute ventilation per kilogram of body weight, viral particles or hazards present in the environment are delivered to children at higher internal doses. Additionally, children often possess less self-control and are more likely to engage in “risk-taking” behaviors [258]. Therefore, children represent an active source of viral transmission, especially to their family, teachers, and playmates [259].

A study [260] has shown that children are exposed to more contaminants than adults due to differences in body size and proportion and breathing zones. For instance, after pesticide application, concentrations of pesticides are always higher closer to the floor, a space often occupied by children. Because children breathe air that is more heavily contaminated due to patterns of evaporation (i.e., re-volatilization) after pesticide application to baseboards, they are exposed to more contaminants than adults. Furthermore, it is important to emphasize that any disruption or damage to a child’s biological/physiological system may be severe and persist as they grow and mature through adolescence and adulthood [260].

Children hospitalized with a coronavirus infection (i.e., SARS, MERS, or COVID-19) tend to be less severely affected than the elderly and often exhibit a mild to moderate course of infection with a favorable clinical outcome [261]. So far, child mortality due to COVID-19 is low. However, children still play an important role in virus transmission which makes it important to understanding the dynamics involved. Therefore, it is necessary to include pediatric considerations in clinical trials for therapeutics and vaccine development as well as preventive measures.

Although preventing respiratory disease and infection in children is an important task, there are no specific masks engineered, designed for, or tested in children. Relevant studies on respiratory PPE for children in their daily routine are very rare [252]. Recently, two studies explored modified N95 respirators and face masks for pediatric examining their fit, leakage, comfort, and safety [93,252]. In an early study on the effectiveness of N95 respirators, surgical masks, and cloth face masks, a protective factor was calculated from measurements of particle concentrations outside and inside the masks worn by healthy adults and children between 5 and 11 years of age [173]. Children were much less protected than adults due to fit problems. While there was a high degree of variability in exposure protection for individual children, as a group they still were protected from virus transmission despite poorly fitting masks.

Children should receive general guidance and education on proper mask wearing, specifically to not touch the mask while wearing or doffing it. Children should wear masks in areas with a high density of people. After wearing a mask for a certain amount of time (e.g., 1 h), children should remove the mask to breathe fresh air, and they should be discouraged from wearing a mask during exercise, particularly an N95 respirator. Additionally, mask selection is important, as properly designed and modified masks are necessary to provide the best protection for children considering fit and comfort. Studies performed by Naomi and others show that a child’s perception of face masks is highly influenced by their design and breathability; therefore, face mask wearability could be improved by designing more visually appealing, breathable, and comfortable masks.

Also, the elderly belong to the most vulnerable segment of our population, as seen by their highest mortality rate due to COVID-19 [262,263]. Unlike children, many older individuals are unable to use RPDs, e.g., because they have a pre-existing chronic disease and RPDs may negatively affect their health due to added breathing resistance and discomfort [194]. Additionally, elderly individuals who are not capable of removing their own RPD when needed should avoid highly breathing-resistant RPDs. Discomfort associated with wearing RPDs may not be tolerable for elderly individuals, especially those with breathing difficulties and potential cardiovascular disorders. Therefore, RPDs addressing the specific needs of children and elderly individuals should be designed based on scientific evidence and usage scenarios.

## 4. Numerical Modeling and Simulation of RPD Infection Control

The development and application of models of virus control and prevention advance our understanding of transmission mechanisms [264,265,266]. These models provide a theoretical foundation for infection control and improvements in RPD engineering, including new mask materials, designs, standards, guidelines, and public health policies. Here, we focus on the use of numerical models to understand the transport mechanisms of virus-laden droplets in the environment-mask-human system (Figure 4), with an emphasis on the RPD filtration mechanism.

An infected host ejects virus-laden saliva droplets to the environment which need to be filtered by an RPD to prevent them from adhering to the floor or other surrounding objects, or remain suspended in the air [267]. Virus-laden droplets are filtered by RPDs mainly by diffusion, electrostatic attraction, interception, inertial impaction, and gravitational sedimentation [268]. While smaller droplets (<1 µm) usually adhere to the mask fibers, larger droplets (≥1 µm) possess a higher inertia which prevents them from following the air-flow (which flows around the mask fibers) and causes them to collide with the mask fibers (Figure 3). For droplets that escape an RPD, Brownian motion plays a major role in the transport and dispersion of small aerosolized particles (<0.5 μm) but becomes less important with increasing particle size [113]. For droplets with diameters >100 μm gravitational settling becomes important and they usually deposit on the ground in liquid form within 1 s. Droplets with diameters <100 μm tend to evaporate, thus leaving behind aerosolized droplet nuclei that can remain suspended for a long time and travel great distances [269].

### 4.1. Simulation of Human Breathing, Talking, Sneezing, and Coughing

The rate of expulsion of respiratory droplets from the mouth and nose can be expressed as the product of the volume expiration rate and the number of droplets per unit volume of exhaled gas [270]. The volume expiration rate of breathing ranges from ~100 mL/s at rest to ~2000 mL/s during intense exercise [271], and the number of droplets per unit volume ranges from ~0.1 to 1 mL [137]. Although the loudness of talking can affect the number of droplets, their diameter typically averages 1 μm when breathing or talking regardless of the loudness [272]. However, coughing and sneezing generate droplets that are >50 μm in diameter. A single cough can produce ~3000 droplets, whereas a sneeze can release an estimated 40,000 droplets [273].

The breathing cycle, talking, sneezing, and coughing can be modeled as using time-dependent velocity profiles for particles ejected through the respiratory airways (i.e., mouth and/or nose) [67,91,118,274]. The exhalation flowrate over time can be represented by a sinusoid for breathing and as a constant for speaking [275]. Eqs. 1 to 3 represent examples of models for shallow, medium, and deep breathing [276]. The sneezing pressure response resembles that of coughing, and both can be modeled using a gamma distribution [116,277]. Different methods to model the coughing and sneezing behavior have been developed by Feng et al. who adopted a Fourier series to characterize the volumetric flow rate at the mouth during a cough [66]. The RPD acts as a viscous and inertial resistance to breathing, sneezing, and coughing causing the pressure drop. The effect of RPD on transmission of virus-laden droplets, trajectories of droplets, and social distancing measures were investigated by researchers in the case of human breathing, talking, sneezing, and coughing [275,276,277,278].

Shallow breath:(1)$$v=0.673t^{3}-4.9557t^{2}+8.242t-0.2621$$

Medium breath:(2)$$v=1.346t^{3}-9.9114t^{2}+16.485t-0.5243$$

Deep breath:(3)$$v=4.0384t^{3}-29.734t^{2}+49.456t-1.5728$$

The size distribution of saliva droplets was initially modeled using a Rosin–Rammler distribution (i.e., Weibull’s law) [67,118,279], which utilizes an equation with the following form:(4)$$f=\frac{n}{\bar{d_{p}}}{\left(\frac{d_{p}}{\bar{d_{p}}}\right)}^{n-1}e^{-{\left(d/d_{p}\right)}^{n},\;\;\;n=8,\;\;\;\;\bar{d_{p}}=80\mu m}$$
where $d_{p}$ is the droplet diameter.

However, sneezing and coughing not only lead to droplet ejection but also generate a multi-phase turbulent gas cloud that can entrain clusters of droplets with a continuum of droplet sizes into the ambient air [280]. In this manner, virus-laden saliva droplets can remain suspended in the air for longer. Therefore, simulating only isolated droplets does not reflect most real-life scenarios.

### 4.2. Simulation of Droplet Transportation and Air Flow

Virus-laden fluid particles include both mucous salivary droplets and multiphase turbulent gas clouds [280]; the latter can be difficult to simulate with high accuracy due to the larger spatial scales involved. The Euler–Lagrange approach can be used to address this problem as it is commonly used to simulate the dynamics of multiphase flows [66,67,116,118,279]. This approach is based on a statistical description of the dispersed phase in terms of a stochastic point process coupled with an Eulerian statistical representation of the carrier fluid phase [281]. To simulate the dynamics of virus-laden fluid particles, the fluid phase (i.e., dry air and water vapor) is treated as a continuum governed by the Navier–Stokes and mass conservation equations, whereas large droplets from sneezing or coughing are treated as the dispersed phase, with individual droplet trajectories predicted using Lagrangian particle tracking (Figure 5). The fluid and dispersed phases are coupled by exchanging momentum, mass, and energy in the calculated flow field.

An alternative approach to simulate multiphase flows is via the Langevin stochastic differential equation of statistical mechanics [128]. This approach considers the importance of diffusion for the motion of small droplets in air, which is often neglected in the Eulerian–Lagrangian approach [118]. However, the Euler–Lagrange approach can be extended to include stochastic models such as the Langevin model to better account for diffusion.

To simulate turbulent airflow, some researchers employ incompressible Reynolds-averaged Navier–Stokes equations with specific turbulence closure models, such as the transition shear stress transport (SST) model [66], the k-ω turbulence model [67,118], the k-ω SST turbulence model [282], the re-normalization group k-ε model [283], and the realizable k-ε model [116]. Some researchers also use the more computationally expensive large eddy simulations to describe the coherent structures of eddies in turbulent air flow [279]. In a study investigating the dynamics of cough and sneeze flows with Reynolds numbers <104, the more accurate but computationally more expensive point particle direct numerical simulation (PP-DNS) method has been used to compute the spread and trajectories of the cough/sneeze flows [117]. The researchers explored the mass loss (evaporation) mechanisms of respiratory droplet on different types of RPD surfaces (permeable and impermeable), which highlighted that the permeable materials are less susceptible to virus survival compared to the impermeable surface [284,285]. Setti el al. (2020) postulated that particulate matters (PM) act as a droplet carrier and could trigger the spread of the virus [286]. Therefore, the penetration and deposition of PM on RPD needs to be analyzed in detail.

### 4.3. Simulation of Particle Transport through RPDs

Some studies treated the RPD fabric as a porous material and adopted the Euler–Euler multi-fluid approach to describe the detailed interaction between virus-laden droplets and the porous material. Willeke et al. [287] investigated how different bacterial shapes, aerodynamic sizes, and flow rates affect penetration through a surgical mask and dust/mist respirator. They found that in size ranges from 0.9 to 1.7 μm spherical corn oil particles and spherical bacteria penetrate similarly, whereas rod-shaped bacteria had less penetration depending on their aspect ratio. Yi et al. [91] developed a one-dimensional mathematical model to describe the mechanisms of virus diffusion in a facemask during the breathing cycle considering vapor molecular diffusion, evaporation/condensation, and sorption/desorption of moisture by the mask fibers, virus filtration, Brownian diffusion of the virus, capillary (wet) virus penetration, and latent heat generation due to phase change.

In other research, the filtration efficiency of an RPD was estimated based on either experimental measurements or on modeling results without explicitly considering the transmission of virus-laden droplets through the internal structure of the RPD material. The emphasis of these studies was on investigating the droplet dynamics as well as their leakage due to poor face seals. The expiratory events increase the pressure and push the face mask outward that result in increased perimeter leakage, such fluid-structure interaction effect was studied by numerical modeling [288,289,290]. Various computational fluid dynamics (CFD) simulations have been performed to investigate the degree to which RPDs reduce droplet travel distance. Feng et al. defined filtration efficiency as a function of particle diameter [66]. Khosronejad et al. modeled face mask filtration efficiency by applying a drag force to the grid nodes used to discretize the three-dimensional geometry of the mask [291]. Dbouk et al. developed a more detailed method for modeling transition modes (i.e., stick, splash/rebound, and penetrate) of a droplet impacting a porous mask filter. In this study, the interaction between droplets and the mask was considered as a function of critical values of the droplet Weber and Laplace numbers, droplet diameter, and splash kinetic energy [67].

More recently, deep learning has been used to analyze the three-dimensional (3D) distribution of particles in N95 respirator filter layers [292]. The 3D internal structure of the filter layer and adhesion of particles to its fibers were characterized by X-ray microscopy. Then, deep learning was used to segment and analyze the 3D data sets containing over 1000 slices for each filter layer, allowing them to quantify the porosity, fiber distribution, and particle distribution in various filter layers. This study showed that fiber diameter and distribution play an important role in filtration efficiency. Regions with higher fiber densities provided a higher filtration efficiency which could be improved even more by using fiber diameters <1.8 μm.

Furthermore, CFD simulations have shown that wearing an N95 respirator results in CO_2_ accumulation as well as increased temperature and pressure inside the respirator microclimate [274]. Examining face seal leakage, another study showed that most leaks appeared near the nose (40%) and cheeks (52%) [289].

### 4.4. Knowledge Gaps and Recommendations for Future Research Directions

From the many approaches that have been used to investigate the effects of human behavior and environmental conditions on face mask performance (Table 5), CFD appears to be the most powerful tool for investigating virus-laden droplet transfer. However, the approach to model the filtering mechanism and breathing resistance of RPDs is often oversimplified, as it neglects virus-laden droplet transmission in the internal RPD structure or elevated CO_2_ levels in the RPD’s dead space and the potentially adverse effects on human physiology. Proper modeling of RPD filtering and leakage caused by fit issues under various application scenarios and human activities is critical both from an engineering and RPD design point of view. In particular, there is an urgent need to understand the interactions between virus-laden droplets and the internal structure and configuration of RPD material to improve protection and wearer comfort.

More realistic simulations of coughing and sneezing are needed to account for facial muscle movement and subsequent changes in air gap size or degree of leakage. The effect of fit on filtration efficiency is also important and should be investigated in greater detail.

**Table 5 polymers-13-04165-t005:** Past research using modeling and other simulation approaches to link human behavior, face mask filtration mechanisms and efficiency, and environmental conditions.

Research Focus		References
Realistic simulation of expiratory events	Coughing	[67,117,118,291]
	Sneezing	[116,293]
Human activities	Head movement	[116,279]
	Walking	[282]
Accurate representation of the interaction between droplets and the RPD’s internal structure		[91,284,285,287,292]
Leakage flows		[66,67,279,290,291,294]
Representation of CO_2_ levels		[212,274]
Validation with experimental data		[116,295]
Effects of environmental conditions	Wind speed	[66,118,291]
	Relatively humidity/particulate matters	[66,286]
	Room ventilation	[113,278,279,296]

## 5. Decontamination and Reuse of RPDs

Despite increased production, the shortage of PPE for healthcare workers and the general public remains an urgent issue in the US [297,298]. The demand for PPE is expected to remain high, with an estimated annual increase of 20% between 2020 and 2025 [299]. The COVID-19 pandemic has worsened the situation with substantially increased demand, leading to further concerns about the environmental impact of discarded RPDs.

Plastic and medical waste is an important environmental issue that has economic, social, and technological aspects [44,45]. Drastically increased demand and usage of medical PPE and supplies have transformed the dynamics of plastic waste generation. A recent study estimated that globally, 129 billion disposable masks and 65 billion disposable gloves are used every month [300,301]. Medical waste is expected to grow by 370% and packaging plastic by 40% in some areas compared to pre-pandemic consumption. (Figure 6) [298,301]. The Chinese city of Wuhan generated nearly 247 tons of medical waste per day at the peak of the COVID-19 pandemic, nearly six times more than before the pandemic [302]. Mixed plastics, like those in single-use masks and other medical supply materials, also pose a great threat to the environment due to their low recyclability [44].

Unsafe disposal of healthcare waste is not only a threat to the environment but also a potential source of disease spread, including hepatitis, HIV/AIDS, and SARS [303,304]. Reuse of disposable medical equipment may also cause respiratory disease [304]. Therefore, adopting advanced waste management, transitioning toward environmentally friendly materials such as bioplastics, and utilizing new sustainable technologies are crucial to fighting future pandemics. Presently, developing a generalizable approach to RPD decontamination and reuse appears promising to mitigate environmental issues in the long term.

### 5.1. Challenges in RPD Decontamination

Understanding virus viability is crucial for developing RPD decontamination approaches. Researchers found that up to 99.8% of respiratory pathogens could be trapped on the respirator after usage or after a simulated cough or sneeze [50,51,305,306,307]. These trapped pathogens [308,309] remain infectious for extended periods of time which poses a serious threat to healthcare workers, as the virus can be further transmitted or re-suspended [50,307].

The viability of SARS-CoV-1 and SARS-CoV-2 has been evaluated on various materials including cardboard, wood, plastic, fabric, paper, glass, rubber, and metal [48,49,310,311,312,313,314]. Virus reduction can take hours to days depending on the material. Some studies measured the viability of SARS-CoV-2 directly on PPE or PPE-derived materials [49,313,315] and found that the virus can remain live and infectious for up to 3 weeks on plastic face shields, N100 respirators, and polyethylene coveralls [49].

The viability of SARS-CoV-2 is also strongly associated with viral load and inoculum size [316]. Its inactivation time has been found to increase from minutes to almost a day with a 100-fold increase in viral load. Furthermore, environmental conditions such as temperature, humidity, and pH also impact virus viability [247]. While SARS-CoV-2 viability declines rapidly under elevated temperatures, it has been shown to remain viable for up to 14 days at room temperature (20–20 °C) [317,318,319]. As humidity is increased, virus survival time first decreases and then increases again, following a “U” shape [296].

Based on virus viability research, the CDC and other researchers have suggested a “round robin rotation” system to reduce the risk of contact transfer of pathogens during FFR reuse [46,320]. In this strategy, each healthcare worker caring for suspected or confirmed COVID-19 patients is issued five N95 respirators, one for each day of the working week. After use, the respirator is stored in a breathable paper bag for at least five days before reuse. This is expected to provide time for pathogens on the respirator to “die off” [46]. However, this strategy has not been validated with FFRs used in real hospital operations.

Decontamination, a process to reduce or eliminate pathogens on a surface, is a strategy for ensuring uninterrupted availability and reuse of RPDs. Although most masks and respirators are designed for single use, several decontamination methods to inactivate viruses and bacteria while maintaining RPD protection and wearablility have been evaluated [321,322,323,324]. According to Heimbuch (2011), the ideal decontamination method will “preserve performance and fit, leave no residual toxicity, and be fast-acting, inexpensive, and readily available” [325]. To assess the effectiveness of different decontamination methods, several measures have been considered including effectiveness against viruses (i.e., reduction in pathogen load), performance preservation (i.e., filtration efficiency, airflow resistance, and physical properties), fit preservation, residual toxicity levels, feasibility of the approach (i.e., fast-acting, cost, and availability), clinical practice outcomes (i.e., infection rate change), and adverse effects experienced by wearers (e.g., skin irritation) [321,323,325].

Many decontamination methods have been applied to N95-like FFRs and surgical masks because they are widely used in clinical settings (Table 6). According to NIOSH and FDA guidelines, several methods could be considered to decontaminate FFRs [36,306], including energetic (e.g., ultraviolet (UV) light, dry and moist heat, microwave-generated steam) and chemical (e.g., alcohol, ethylene oxide, bleach, vaporized hydrogen peroxide (H_2_O_2_)). The most popular decontamination method explored is UV germicidal irradiation (UVGI), including ultraviolet C (UVC) irradiation [46,326,327]. UVC irradiation is strongly absorbed by RNA and DNA bases, leading to molecular structural damage and protein denaturation, which results in virus inactivation [328,329]. Energetic methods rely on various types of heating that induce structural changes in virus proteins, disrupting the specific structures needed to identify and bind to host cells [330]. Chemical methods include several disinfection mechanisms, including cross-linking, coagulating, and clumping; structure and function disruption; and oxidation [331,332]. For example, alcohol causes cell proteins to clump and become denatured resulting in the collapse of cell membranes and cell death. Chlorine oxidizes proteins, lipids, and carbohydrates. Hypochlorous acid, formed by dissolving chlorine in water, targets key metabolic enzymes and can destroy the organism. Chemical disinfection is more suitable for hard surfaces and specific PPE items such as goggles, but may not be suitable for FFRs because the post-treatment (e.g., rinsing) may lead to undesirable side effects such as a filtration performance decrease [320,333,334], residual chemical toxicity, or odor [335].

From among the tested decontamination methods for FFRs, UVGI, vaporized H_2_O_2_ (VHP), and moist and steam heat decontamination have been proven to be the most effective [46] for a range of pathogens, namely H1N1, H5N1, MS2, and SARS-CoV-2 [47,325,336,337]. For SARS-CoV-2, several studies have shown that a UVGI dose of about 18 kJ m-2 can effectively disinfect contaminated FFRs while preserving their filtration efficiency and fit performance [325,336,338]. In contrast, current findings on the impact of VHP on filtration efficiency and fit are still contradictory [47,333,339,340]. While moist heat has been shown to be an effective method for reducing RPD virus loads [337,341], it may affect fit, odor detection, comfort, or donning difficulty [335].

Even with proper decontamination, restrictions limit the number of times an FFR can be reused [339,342]. While N95 respirators are considered apt for “limited reuse”, there is currently no method to reliably determine the maximum possible number of safe reuses that would hold in all usage scenarios [343]. Nevertheless, for specific viruses like SARS-CoV-2, a conservative reuse range can be determined for specific conditions. For instance, the CDC recommended to limit the number of reuses to five per device based on published data on changes in FFR structure, fit, and performance after multiple donning [46]. However, research has shown that filter efficiency and fit could be maintained through ten UVGI cycles [344], while other methods resulted in reduced reusability (three cycles with moist heat treatment [345], five with dry heat [342], and three with VHP [47]). However, results still vary considerably for different decontamination methods. For example, microwave-generated steam decontamination methods preserve respirator fit and function even after twenty cycles [337], whereas ethylene oxide gas, liquid H_2_O_2_, and bleach methods can change FFR filtration efficiency after only one cycle [346]. The number of reuses can be affected by the FFR type, decontamination method, and use intensity and practice. More research is needed to investigate the effects of different decontamination methods on a broad range of FFR types.

Apart from effectiveness, feasibility and potential adverse side effects must also be considered before deciding on a specific decontamination method. Cycle time, accommodation (e.g., cost, accessibility, and acquisition of materials), and residual toxicity are the most common concerns [321,325] and must be adequately addressed. Although UVC is widely applied, its effectiveness depends on the dose and duration which may not be the same for different FFR types [344,347,348]. The UVGI dose is critical: while an insufficient dose may not reach all interior layers and thus leave behind some active infectious agents, too high a dose could affect the structural integrity of the FFR and thereby reduce its filtration performance [349]. It is not sufficient that a decontamination method maintains a good FFR performance, it must also be feasible to integrate it into a clinical work flow. For instance, steam, UVGI, and dry heat, while effective for decontamination, can potentially melt the metal components of respirators [347]. Some methods require substantial processing times, specialized equipment, operator training, and adequate ventilation, e.g., those involving vaporized or liquid H_2_O_2_ or ethylene oxide gas [46]. Microwave ovens, despite their convenience and accessibility, can only treat one FFR at a time [325]. Also, the metal nosebands of some FFRs will cause arcing or sparks during microwave treatment [348].

**Table 6 polymers-13-04165-t006:** Summary of the most commonly used respirator decontamination methods [47,322,323,325,327,335,336,339,340,341,344,347,350,351,352,353,354,355,356,357].

Decontamination Method	Disinfection Method	Anti-Pathogen and Performance Impact	Feasibility and Limitations
Energetic Various types of heating to induce structural changes in virus proteins and disrupt the specific structures needed to recognize and bind to host cells	Ultraviolet germicidal irradiation (UVGI)	Effective against SARS-CoV-2 Usable for multiple decontamination cycles Filter efficiency and fit maintained	Short treatment time Specialized equipment and training needed Dedicated space required Not suitable for home use Appropriate radiation intensity needed
	Moist heat	Effective against SARS-CoV-2 Usable for multiple decontamination cycles Filter efficiency and fit may be altered	Short treatment time Easy to use Low cost Easily available equipment Appropriate heat and humidity needed
Chemical To induce cross-linkage, coagulation, and clumping thereby disrupting the structure and function, affecting or killing an organism via oxidation	Vaporized H_2_O_2_ (VHP)	Effective against SARS-CoV-2 Usable for multiple decontamination cycles Filter efficiency and fit may be altered	Long treatment time Dedicated space required Residual odor Specialized equipment and training needed
	Home bleach	No adept evidence against SARS-CoV-2 Restricted numbers of decontamination cycles Filter efficiency and fit may alter	Easily available equipment Easy to use Effectiveness may vary Residual odor

### 5.2. Surgical and Cloth Mask Decontamination

Existing research on decontamination methods for surgical masks is limited [323,324,358,359,360]. One study examined the effectiveness of dry heat (using a rice cooker), high-pressure moist heat (autoclave), and three chemical agents (70% ethanol, 100% isopropanol, and 0.5% sodium hypochlorite/bleach) on N95, gauze, and spunlace masks [360]. All five methods reduced the masks’ particulate filtration efficiency, with bleach having the largest effect and dry heat the smallest [360]. Another study investigated the feasibility of several decontamination methods for surgical masks [324] and found that aqueous H_2_O_2_, aqueous sodium hypochlorite, and ultraviolet irradiation maintained the filtration efficiency while effectively disinfecting from the A-H1N1 virus. Furthermore, mask performance was still acceptable after 10 cycles of H_2_O_2_ or diluted bleach treatment or 30 cycles of ultraviolet irradiation treatment [324]. Hence, ultraviolet irradiation and chemical methods appear to be effective approaches to decontaminate surgical masks, although further studies are needed for other mask types and materials.

So far, guidance and research related to cloth mask decontamination is very limited due to their wide range of materials and designs [361,362,363,364]. While many decontamination methods recommended for FFRs and surgical masks may not be appropriate for cloth masks, mostly due to resource and feasibility limitations, some can be easily implemented as the necessary equipment is widely available, e.g., dry or moist heat can be applied with a rice cooker or kitchen steamer [365]. Using MS2 as a surrogate for SARS-CoV-2, one study showed that moist heat could sufficiently inactivate the virus, outperforming dry heat at the same temperature [365]. A newly developed method using hot hygroscopic materials also showed positive cloth mask decontamination outcomes [366]. Additionally, hand or machine washing with bleach are widely adopted by the general population [361]. However, cloth mask performance has not been evaluated for these treatment types. Thus, additional studies are needed to investigate the factors that affect cloth mask decontamination effectiveness and reuse frequency for the available decontamination methods and specific settings.

### 5.3. Current RPD Decontamination Regulations and Issues

Several guidelines for RPD decontamination and reuse have been issued by the CDC, OSHA, and FDA in an attempt to remedy RPD shortages during the COVID-19 pandemic [46,367,368]. In particular, the FDA issued an emergency use authorization for decontaminating certain disposable surgical masks and FFRs. Given the unclear consequences for FFR performance, NIOSH suggested to use only the following methods: UVGI, VHP, and/or moist heat [46]. If neither of these methods are available, microwave-generated steam and/or liquid H_2_O_2_ can also be applied. Importantly, the following methods should not be used until research has been able to demonstrate their safety and effectiveness: dry heat (e.g., autoclaving), isopropyl alcohol, soap, dry microwave irradiation, chlorine bleach, disinfectant wipes, and ethylene oxide gas.

For all decontamination methods, however, additional research is needed to further validate their efficacy and safety prior to implementation. In particular, application settings, standard procedures, and targeted RPD types must be clearly specified. Additionally, much research has focused on decontaminating the actual facepiece of an FFR but accessories such as straps have been largely ignored [369]. It is critical to ensure the intactness of the entire FFR to be able to guarantee safe usage. Moreover, as the efficacy of most decontamination methods has been tested with surrogate viruses, such as influenza A, many methods still need to be tested with the actual SARS-CoV-2 virus. It is also questionable whether the artificial virus inoculation employed by many existing studies is a fair representation of actual mask contamination. Thus, better virus contamination simulation is needed for improved realism and reliability of the results. Furthermore, treatment cycle time and intensity should be optimized specifically for use against SARS-CoV-2 while ensuring that the performance and intactness of FFRs and masks are maintained. Although these methods provide evidence of bio-decontamination, soil, dirt, and trapped particles on FFRs also need to be properly dealt with to ensure adequate hygiene and good usability. The overall lack of understanding of how surgical and cloth masks should best decontaminated warrants further research.

## 6. New Materials for Future RPDs

### 6.1. Nanomaterials

The size of pathogens, including bacteria and viruses, ranges from tens of nanometers to several micrometers. Densely packed porous materials with electrostatically charged fibers are crucial for efficient pathogen filtration while also allowing sufficient air permeation. Nanofibrous membranes (NFMs) are constructed using multilayers of randomly distributed nanofibers, whose diameters range from a few hundred nanometers to several micrometers and the pore sizes are in the micrometer range. They have been developed to effectively filter small particles such as PM2.5 and microorganisms based on size-exclusion [370,371,372]. The filtration efficiency is highly dependent on fiber diameter, pore size, and NFM thickness [370,373,374,375]. With the acute shortage of medical face masks during the early stages of the COVID-19 pandemic, many studies began to investigate the filtration efficiency of various conventional textiles for use as face coverings, finding a large amount of variability based on textile type, number of layers, thickness, and weaving density [77,376].

Rough and hairy surfaces of textile materials were found to improve filtration efficiency, which led to the introduction of rough morphological structures into nanofibrous materials to increase the specific surface area and porosity of NFMs and thereby increase the probability of capturing smaller particles such as viruses. Examples of NFMs containing a so-called “valley-and-rib” structure [377], secondary meso- and micropores [378,379], nanonets [380,381], and other hierarchical nanostructures (e.g., nanoparticles, nanorods, nanoflakes) [382,383,384] are shown in Figure 7a. The addition of nano-roughness [385,386] and hydrophobic properties [387,388] to fiber surfaces can increase pathogen repulsion by creating a superhydrophobic-antifouling feature (Figure 7b), mimicking the surface of lotus leaves.

The surface chemistry of NFMs is also essential to filtration performance, especially of biomolecules [389]. Interfacial interactions between pathogens and filtering materials can be controlled to repel or attract pathogens [390] instead of relying solely on size-exclusive filtration. Cationic polymers are fabricated to enhance the adsorption of negatively charged pathogens via electrostatic interaction, thus sequestering pathogens on fiber surfaces and reducing their penetration and transmission [391]. Abundant functional groups on chemically modified surfaces of nanofibers could further increase selective adsorption of the biomolecules [392,393]. In addition, selective binding of viruses on nanofibrous materials can be achieved via biological recognition by incorporating anti- or nanobodies into the surfaces of nanofibers [394,395]. Such biological interactions could also improve the protective performance of the filtering media by selectively capturing pathogens on fiber surfaces. Moreover, applicable photocatalytic oxidation through a nanocoating on textiles also provides decontaminating functions, especially in an indoor environment [396].

### 6.2. Biodegradable Materials

Current N95 respirators and surgical face masks are primarily made of olefin polymers, which are inert and persistent in the environment. Biodegradable face masks and respirators are highly desired to lessen the environmental burden posed by existing PPE materials. Attempts to construct RPDs using nonwoven fabrics by meltblowing biodegradable polylactide and polyesteramide resulted in materials with comparable mechanical properties as conventional PPEs [397]. Electrospinning is another approach to fabricate effective filtering layers for face masks and respirators from biodegradable polymers, including polysaccharides (e.g., cellulose acetate, chitosan, starch, alginate), proteins (e.g., gelatin, silk fibroin), and synthetic polymers (e.g., polylactic acid, polyvinyl alcohol (PVA), polycaprolactone, and polyethylene glycol) [268]. The nanofibrous membranes can replace meltblown fabrics in the SMS structures of RPDs, while spunbond layers are necessary to provide the desired mechanical strength and stability. While most natural or biodegradable polymers may not be suitable for electrospinning into NFMs, the addition of a second polymer component (e.g., polyethylene glycol) could improve their spinnability and mechanical strength [398,399,400,401,402,403].

### 6.3. Biocidal Materials

Bacteria and viruses can survive on many surfaces for days and even weeks, maintaining their infectivity and thus causing further transmissions. In fact, the filtering performance of RPDs can concentrate pathogens on the outer layer of N95 respirators and surgical face masks. Adding pathogen-killing properties to face masks and RPDs could provide improved protection by reducing transmissions and cross-infections during PPE doffing and disposal. Any pathogen-killing functionality (i.e., biocidal function [404]) in mask materials should be safe to wearers, rapidly inactivate viruses and bacteria upon contact, and be stable for prolonged use. Only few biocides meet these general requirements and are mostly disinfectants with only limited alternative options [405,406]. The fact that the biocide comes into close contact with human tissue and the respiratory system further narrows the range of suitable agents. The few successful approaches to develop antiviral face mask materials are summarized below.

#### 6.3.1. Metal Ions and Nanoparticles

Several metals and related derivative compounds (e.g., copper, silver, and their nanoparticles) possess safe biocidal and antiviral properties. Copper, copper ions, and CuO nanoparticles inactivate bacteria by damaging cell membranes or geno-toxicifying cells; furthermore, they kill viruses by generating reactive oxygen species (ROS) [407]. Silver ions can damage cell membranes and bind with sulfhydryl groups in proteins, which should be effective in inactivating bacteria cells [408,409]. However, all nanoparticles are considered potentially harmful to humans, and their use in face masks may pose serious health risks.

#### 6.3.2. Quaternary Ammonium Salts

Quaternary ammonium salts (QASs) are well-known antibacterial agents used in personal and medical hygiene products. QASs work through a combination of electrostatic adsorption and physical penetration into bacterial cells. Some QASs also display biocidal activity against viruses and are recommended by the U.S. Environmental Protection Agency for disinfection against SARS-CoV-2 [410,411]. However, QASs do not kill microorganisms rapidly, which limits their suitability for RPDs. Nevertheless, their cationic structures may enhance the interactions of microbes with the surface of fibrous materials and thereby achieve a certain level of antibacterial functionality, which has resulted in them being incorporated into certain textile products. Recently, QAS structures have been employed to bond organic photosensitizers onto fiber surfaces which could significantly increase contact of bacteria and viruses with the agents and lead to their rapid inactivation when exposed to sunlight [412].

#### 6.3.3. Photo-Induced Biocides

Organic photosensitizers (PSs) can effectively produce biocidal reactive oxygen species (ROS), including H_2_O_2_, hydroxyl radical (HO•), superoxide radical (•O_2_–), and singlet oxygen (^1^O_2_), under irradiation by UV or visible light. The ROS production by PS can be explained with the Jablonski diagram (Figure 8a). ROS are lethal to microorganisms as they oxidize proteins, RNA, DNA, and lipids [406]. Unlike conventional antibiotics, killing pathogens with ROS is non-selective and ultra-fast (i.e., within minutes to hours) [412,413,414]. The short lifetime (~µs) and limited diffusibility (nanometer range) of the main ROS (HO• and ^1^O_2_) ensure that their biocidal properties only persist briefly and on targeted surfaces without entering the respiratory systems or affecting the mucous membranes of wearers, which makes this a promising approach for use in PPE and especially face masks (Figure 8b). Commonly used PS can be classified according to their photoactive structures, such as anthraquinone [412,415,416,417,418,419], benzophenone [413,420,421,422,423,424], xanthene [412,425,426,427,428,429], porphyrin [430,431,432], phthalocyanines [433], and BODIPY [434] (Figure 8c). They have been introduced into flexible and wearable substrates to inspire the design of next-generation RPDs and face masks that possess enhanced biological protective functionality and reduce environmental concerns. The incorporation of small molecular PSs onto fibrous materials without affecting their photo-induced biocidal activity can be achieved by conventional dyeing (i.e., physical adsorption by weak bonds or formation of covalent bonds) (Figure 9a) [412,427], chemical modification (Figure 9b) [419,428], co-electrospinning [434], guest-host adsorption (Figure 9c) [429], and layer-by-layer assembly [415,417]. Instead of using single molecules, porphyrin has been introduced as a building block for photoactive metal organic framework (MOF) synthesis (PCN-224). The antibacterial properties of the cotton fabric are achieved by growing PCN-224 in situ on the fiber surfaces [432].

It should be noted that some anthraquinone and benzophenone derivatives have been reported to be environmental pollutants and endocrine disruptors, which may limit their practical application [435,436]. Moreover, although PSs used in medical treatment (e.g., rose Bengal) possess antimicrobial properties when exposed to light, potential health concerns associated with their daily oral or dermal administration must be resolved before they can be recommended for general use in face masks and other PPE. Thus, bio-based PSs and derivatives with high levels of safety and biocompatibility are very promising for use in face masks. Recently, it has been found that the photoactivity of the vitamin K family, vitamin B_2_, and curcumin can produce sufficient amounts of ROS when exposed to light (Figure 8d) [437]. Photo-induced antimicrobial and antiviral properties were obtained with nanofibrous membranes loaded with vitamin K_3_, polyethylene films containing vitamin B_2_, and materials conventionally “printed” with curcumin [414,438,439].

Similarly to small molecular PSs, conjugated, photo-excitable oligomers and polymers have also exhibited biocidal functionality when exposed to light. Three highly conjugated oligomers and two cationic conjugated polymers with demonstrated photo-induced antiviral activity against SARS-CoV-2 have been synthesized using phenylene ethynylene and polythiophene [440] (Figure 8e). Polythiophene-based polymer-poly [2,11′-thiopheneethylenethiophene-alt-2,5-(3-carboxyl)thiophene] is another example of a polymeric photosensitizer that has shown broad light adsorption across the visible range with an efficient generation of ^1^O_2_ and free electrons, which is promising for an antibacterial coating [441].

**Figure 8 polymers-13-04165-f008:**
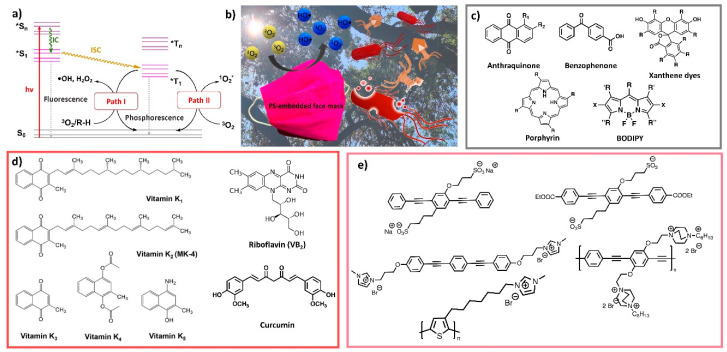
(**a**) Jablonski diagram illustrating the photo-excitation process of PSs. Reprinted with permission from [412] © 2020, American Chemical Society. (**b**) Illustration of biocidal activity of face masks embedded with PS. Reprinted with permission from [412] © 2020, American Chemical Society. Chemical structures of (**c**) synthetic PSs embedded in fibrous materials, (**d**) bio-based or bio-derived PSs. Reprinted with permission from [437] © 2019, American Chemical Society; (**e**) conjugated oligomers and polymers with antiviral functions. Reprinted with permission from [440] © 2020, American Chemical Society.

**Figure 9 polymers-13-04165-f009:**
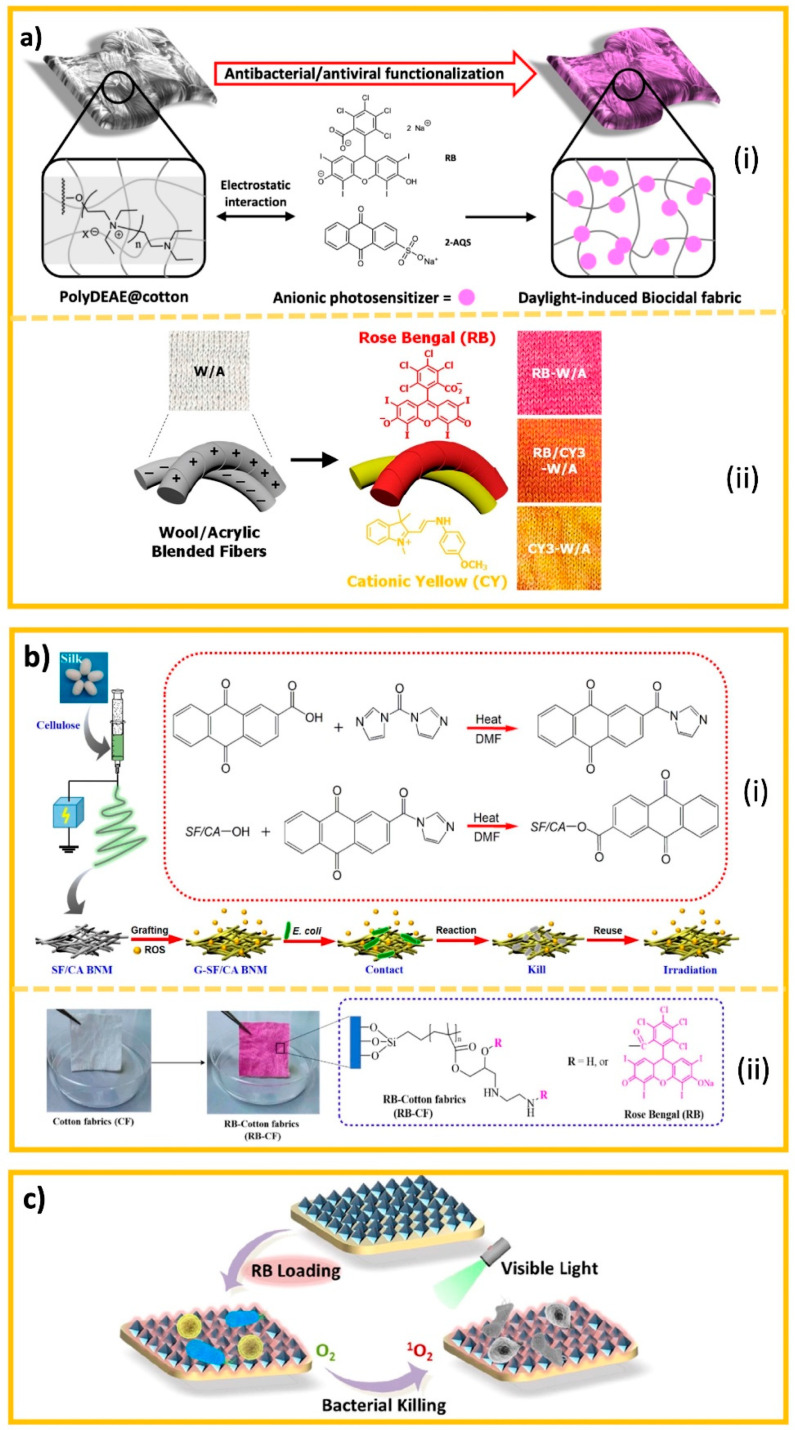
Incorporation of PSs into fibrous materials based on (**a**) conventional dyeing (i.e., electrostatic interaction): (i) photosensitizers dyed on polycationic cotton fabrics (reprinted with permission from [412] © 2020, American Chemical Society), (ii) photosensitizer dyed on wool/acrylic blended fibers (reprinted with permission from [427] © 2019, American Chemical Society); (**b**) chemical modification: (i) chemical grafting of photoactive anthraquinone moiety on silk fibroin/cellulose acetate blend nanofibrous mem-brane (reprinted with permission from [419] © 2020, American Chemical Society), (ii) chemical grafting of rose Bengal on cotton fabrics (reprinted with permission from [428] © 2020, Elsevier); (**c**) guest-host adsorption (reprinted with permission from [429] © 2020, Elsevier).

#### 6.3.4. Halamine Biocides

N-halamine compounds are chemicals possessing imide, amide, and amine N-Cl or N-Br structures. They are powerful biocides that, upon direct contact, can rapidly inactivate large populations of various microorganisms within a few minutes [442]. Halamine structures have been incorporated into fibers, films, fabrics, and polymers to provide chlorine-rechargeable biocidal functions and are widely employed in water disinfection, food safety materials, and biocidal textiles [442,443,444]. However, the release of free chlorine, a strong oxidizer and known biocide [445], can cause irritation, drying, and potentially burns to mucous membranes and lungs. Thus, N-halamine structures are not suitable for face masks and respiratory devices despite their excellent biocidal performance.

## 7. Future Perspectives and Outlook

RPDs, a subcategory of PPE, are used for both respiratory protection and source control. Especially during a pandemic they play a crucial role for human health, safety, and infectious disease control. Future RPDs should be more functional, intelligent, adaptive, sustainable, and offer better protection and comfort through the use of novel materials, sensing technology, advanced engineering, and better ergonomic design. To achieve these goals, we need to improve our understanding of the dynamic interactions between infectious diseases, RPDs, and humans, by employing a broad approach that involves realistic simulations, advanced testing methodologies, refined conformity assessment, and effective surveillance.

First of all, the development of new, innovative materials enabling many desirable functions for future RPDs. Delicate designed nanoscale materials can make various 3D structures to provide sequenced and layered capturing of harmful particles, thereby substantially improving filtration efficiency [377,379,383,412]. Biodegradable materials can significantly reduce the environmental impact of single-use RPD products [268,397]. Additionally, biocidal materials can facilitate self-disinfection and therefore prevent secondary contamination and significantly improve the safety and reusability of RPDs [412,428]. Advances in sensing and wireless technologies can be incorporated into RPD designs for hazard detection, wearer physiological response monitoring, health or disease indication, and interpersonal communication [446,447,448,449]. Such technology can also be used to monitor user-compliance and life-span tracking of RPDs. The realization of cost effective, environmentally friendly, washable, durable, flexible, and scalable e-textiles such as graphene based materials can significantly accelerate the incorporation of wearable technology into future RPDs [450]. Self-powered masks that generate energy through human breathing to power the wearable technologies have also been explored [451]. Hearing-impaired people can benefit from the use of transparent filter material and specific RPD design [370,452]. More excitingly, fibrous materials based on colorimetric sensors are being developed for the detection and signaling of virus existence to warn wearers about RPD contamination [453,454].

Additionally, technological advancements will lead to revolutionary new designs, as well as advances in the development and manufacturing of novel RPDs. In light of the Industry 4.0 concept, many technologies and techniques such as 3D body scanning and 3D printing have been applied to RPD customization and rapid prototyping [455]. For example, using combined polymers, researchers successfully realized stable and biocompatible 3D printed N95 masks that provided a good and comfortable fit for wearers [456]. Reusable masks using 3D printed frames and filter fixtures in combination with replaceable filter media have also been prototyped [457]. A different 3D printed design employs the same concept but with a smaller area of the filter media which significantly increases the life span and efficiency of RPDs while reducing their environmental impact [458]. The ability to adapt and combine existing respirators via 3D printed fixtures is expected to ease the short supply of FFRs during a pandemic [459]. Furthermore, new technologies are being applied to address issues of wearer discomfort. For example, researchers added air-conditioning and air-circulation to RPDs to reduce their breathing resistance and improve thermal comfort [212]. Specifically, researchers have utilized 3D scanning technology to develop customized RPDs that provide a near-perfect fit with a significantly reduced contact pressure, thus enhancing wearer comfort [460].

Other Industry 4.0 technologies such as virtual reality and augmented reality are also being employed to tackle COVID-19 issues [455,461]. These technologies facilitate the creation of virtual clinics with remote consultations, thus reducing the risk of infection and maintaining a functioning healthcare service. These tools can also be employed to educate and train healthcare workers (HCW)s in the proper usage (including donning and doffing) of RPDs in a simulated workspace, thereby preparing them for possible future pandemics.

Moreover, advanced evaluation tools and modeling approaches are currently being developed to facilitate a more comprehensive assessment of RPD performance that considers synergistic effects between different factors such as environmental conditions, hazard nature, usage scenarios, and user behaviors. While preliminary efforts have focused on improved hazard and user behavior simulation [138,141,147], a more systematic tool that can advance our understanding of possible risks and transmission paths is still needed. Such a tool could revolutionize RPD testing, offer holistic information for research and product development purposes, and inform about best practices of RPD usage. Furthermore, it could help to improve existing model parameterizations used for the evaluation of RPD performance in a variety of situations and occupations, thus improving the health and safety of the general public at significantly reduced costs [32,60,72].

The incorporation of artificial intelligence (AI) can further improve existing modelling approaches. Machine learning, especially deep learning, provides novel tools to investigate RPD filtration mechanisms and the performance of RPDs [292]. Numerous deep learning algorithms are being developed for recognizing, detecting, counting, and tracking objects, which can be potentially repurposed to optimize RPD filter efficiency of droplets and aerosols while allowing sufficient airflow. Such algorithms are complementary techniques to simulation-based performance assessments as they can help to investigate the adhesion of contaminant particles to polymer fibers under various environmental conditions. Additionally, AI can also be used to track individuals anonymously and predict outbreaks within a certain geographical region, thus helping administrators and decision makers with the implementation of suitable disease control strategies [462,463,464].

The design of next generation high performance RPDs comes with inevitable trade-offs between factors such as filtration efficiency, fit, breathability, comfort, wearability, decontaminability, and environmental footprint. Multi-objective optimization techniques, e.g., Pareto frontiers [465,466], are useful for designing RPDs for a variety of user groups and usage conditions. A Pareto optimal design cannot be dominated by any objective without compromising another objective. The set of all Pareto optimal designs constitutes the Pareto frontier, which offers a variety of designs that are optimal under different usage scenarios. Identifying Pareto optimal solutions is a combinatorial optimization problem involving multiple decision variables, including material types, technologies, designs, each of which having a large number of possible options. Enumerating all possible combinations will be prohibitively expensive and time consuming. Alternatively, AI powered simulation optimization and active learning algorithms can be used to strategically explore the solution space for near-optimal solutions.

Multi-objective optimization techniques have been used in several studies to address challenges posed by the COVID-19 pandemic. Singh et al. (2020) designed a multi-objective fitness function to classify COVID-19-infected patients by considering sensitivity and specificity [467]. Libotte et al. (2020) proposed a strategy for vaccine administration in the COVID-19 pandemic to minimize both the number of infected individuals and the number of prescribed vaccine doses [468]. Technology advancements improve our understanding of the interactions between hazardous environments, RPDs, and user-specific factors.

New materials and advances in manufacturing have made customized and optimized RPD products real possibility. Next generation RPDs may possess biocidal functions that can be renewed by simple at-home treatment to gain high protection and reusability. The materials used for RPD will largely be biodegradable ones for reducing waste or possibilities to cause pollution. RPDs may also be redesigned to possess much improved wearing comfort with close to perfect fit to sustain extended usage periods and eliminate leakage issue. Additionally, proper training, education, and monitoring of RPD distribution and usage can also be achieved via AI powered algorithms and greatly contribute to public health and lead to societies that are better prepared and therefore capable to remain prosperous while weathering future pandemics.

## Figures and Tables

**Figure 1 polymers-13-04165-f001:**
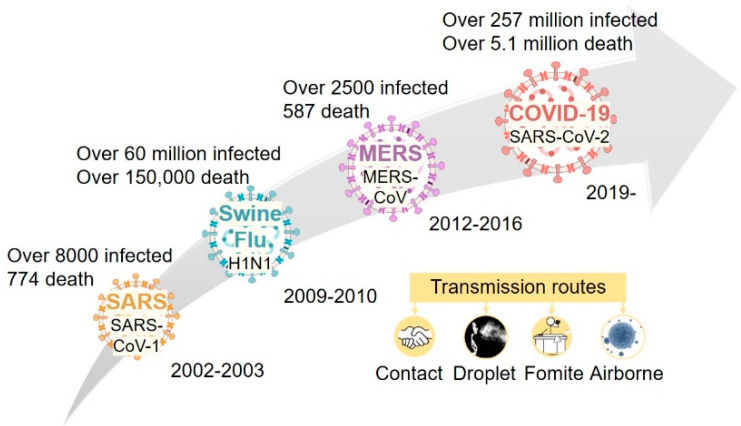
Emerging novel respiratory infectious diseases over the last two decades.

**Figure 2 polymers-13-04165-f002:**
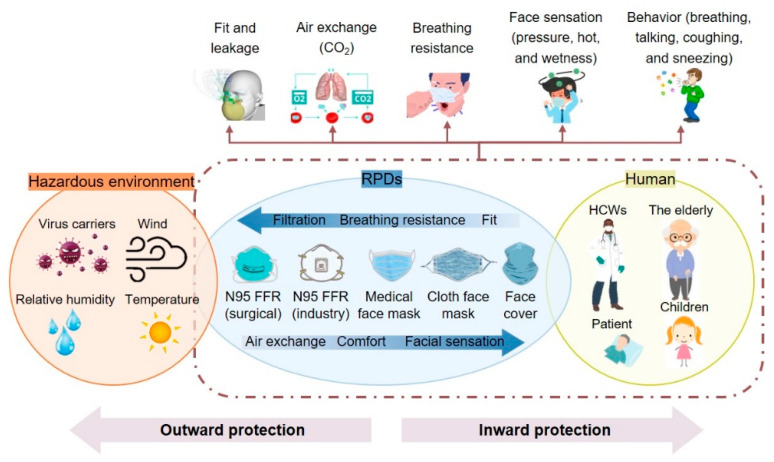
Interacting factors for different hazardous environments, respiratory protective devices (RPDs), and end users and their impact on protective performance and comfort.

**Figure 3 polymers-13-04165-f003:**
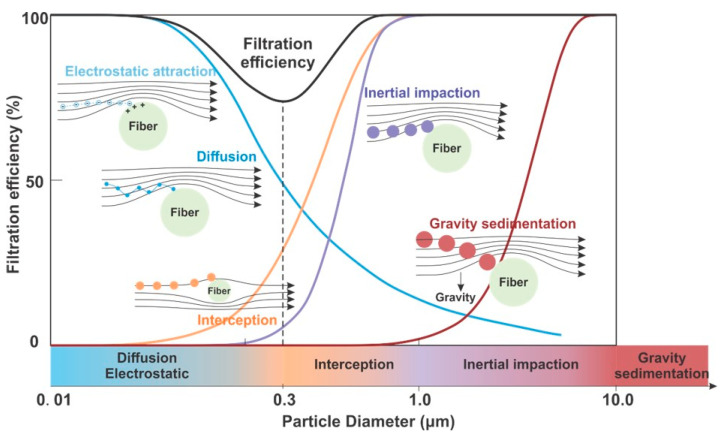
Impacts of particle size and type of filtration mechanism on RPD filtration efficiency.

**Figure 4 polymers-13-04165-f004:**
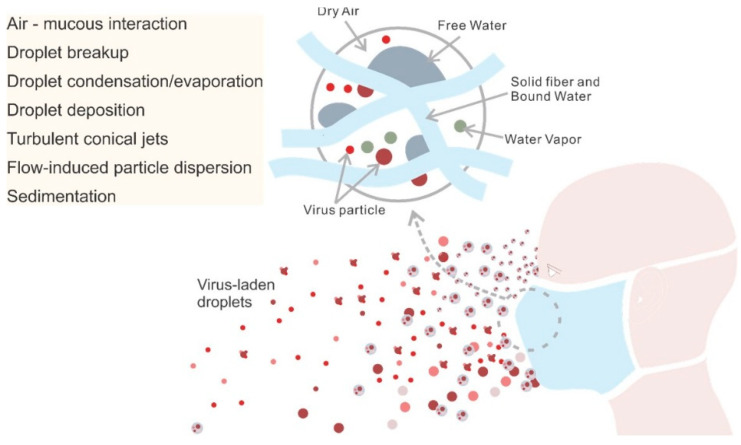
Schematic to illustrate the transport processes of virus-laden droplets in the human-mask-environment system.

**Figure 5 polymers-13-04165-f005:**
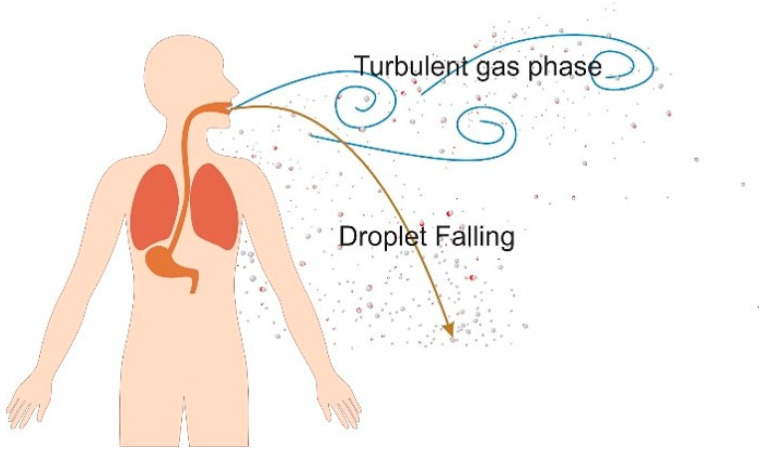
Transport of droplets in the air as a multi-phase flow.

**Figure 6 polymers-13-04165-f006:**
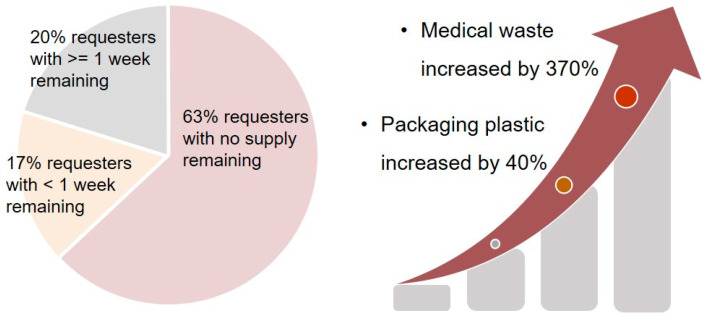
Respirator shortage and medical waste increase as a consequence of COVID-19.

**Figure 7 polymers-13-04165-f007:**
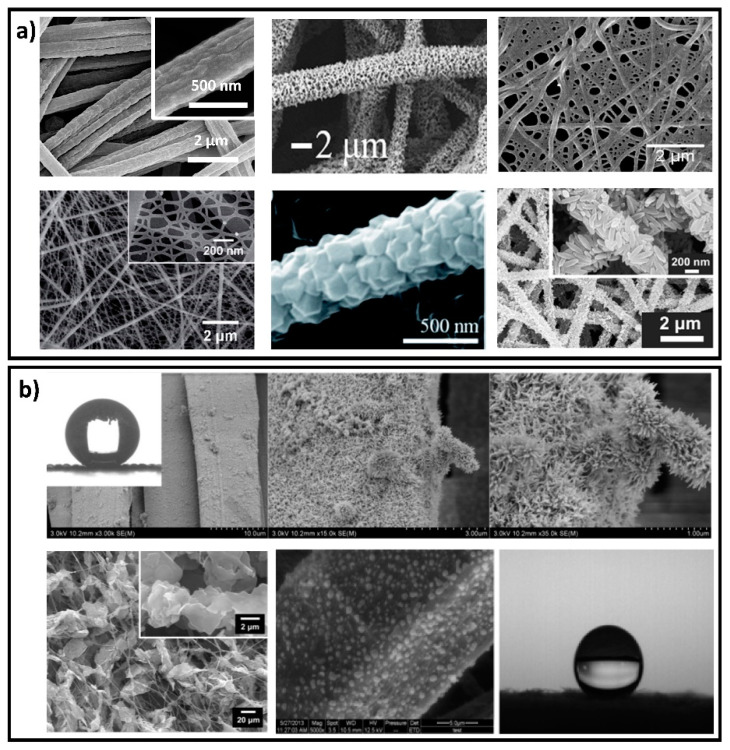
(**a**) Scanning electron microscope images of fibrous materials containing novel secondary nanostructures. Reprinted with permission from [377] © 2019, Wiley; [378] © 2019, American Chemical Society; [380] © 2016, Royal Society of Chemistry; [381] © 2020, American Chemical Society; [382] © 2018, Royal Society of Chemistry; [383] © 2019, American Chemical Society. (**b**) Scanning electron microscope images of superhydrophobic fibrous materials. Reprinted with permission from [385] © 2020, Elsevier; [386] © 2020, American Chemical Society; [388] © 2017, Elsevier.

**Table 1 polymers-13-04165-t001:** Environmental and human factors that influence RPD filtration efficiency.

Influencing Factors	Filtration Efficiency Change	Reference
Particle size	With decreasing particle size, the filtration efficiency starts to decrease before increasing again (Figure 3)	[93,94]
Airflow rate/face velocity	The higher the airflow rate, the lower the filtration efficiency	[77,84,95]
Breathing pattern	Unsteady breathing pattern reduces filtration efficiency	[79,84,87]
Respiration frequency	Higher respiration frequency reduces filtration efficiency	[79,86]
Humidity	Higher humidity reduces filtration efficiency	[84,90,96]
Loading time	Longer loading time increases filtration efficiency	[79,89]

**Table 2 polymers-13-04165-t002:** FFR regulations, specifications, and test methods [54,97,98,99,100].

Specified Performance	United States		European Union	
	Regulation/Guidance: OSHA 29 CFR 1910.134 NIOSH 42 CFR 84		Regulation/Guidance: EU 2016/425 PPE Regulation EU 2017/745 Medical Device Regulation	
	Requirements: NIOSH 42 CFR 84	Test Methods	Requirements: EN 149 +A1	Test Methods
Particulates filtration efficiency (%)	N95 R95 P95 ≥ 95 N99 R99 P99 ≥ 99 N100 R100 P100 ≥ 99.97	TEB-APR-STP-0051 to 0059 Challenge with NaCl for N series; Dioctyl phthalate for R and P series; Flow rate 85 L/min; 0.075 µm CMD with GSD < 1.86; <200 mg/m^3^	FFP1 ≥ 80 FFP2 ≥ 94 FFP3 ≥ 99	EN 149 +A1 EN 13274-7 Challenge with NaCl and paraffin oil; Flow rate 95 L/min; 0.06–0.1 µm CMD with GSD 2–3; 8 ± 4 mg/m^3^
Total inward leakage (TIL, %)	NA	NA	FFP1 ≤ 22 FFP2 ≤ 8 FFP3 ≤ 2	EN 149 +A1 EN 13274-1 Challenge with NaCl on human subjects
Breathing resistance (inhalation)	All N, R, P series ≤ 35 mm H_2_O (343 Pa)	TEB-APR-STP-0007 Head form manikin test; Flow rate 85 L/min	FFP1 ≤ 0.6 (mbar, 60 Pa) and 2.1 (210) FFP2 ≤ 0.7 (70) and 2.4 (240) FFP3 ≤ 1 (100) and 3 (300)	EN 149 +A1 EN 13274-3 Head form manikin test; Flow rate 30 L/min and 95 L/min
Breathing resistance (exhalation)	All N, R, P series ≤ 25 mm H_2_O (245 Pa)	TEB-APR-STP-0003 Head form manikin test; Flow rate 85 L/min	FFP 1, 2, 3 ≤ 3 mbar (300 Pa)	EN 149 +A1 EN 13274-3 Head form manikin test; Flow rate 160 L/min
Exhalation valve leakage	Leakage ≤ 30 mL/min	TEB-APR-STP-0004 At −22 mm H_2_O (−245 pa)	NA	NA
CO_2_ content requirement (%)	NA	NA	FFP 1, 2, 3 ≤ 1	EN 149 +A1 EN 13274-6 Head form manikin test
Flammability	NA	NA	Pass	EN 149 +A1 EN 13274-4
Biocompatibility	NA	NA	Pass	ISO 10993-1 ISO 10993-5 ISO 10993-10

**Table 3 polymers-13-04165-t003:** Surgical/medical mask regulations, specifications, and test methods [97,100,101,102,103,104,105,106,107,108,109].

Specified Performance	United States		European Union	
	Regulation/Guidance: OSHA 29 CFR 1910.134 OSHA 29 CFR 1910.1030 FDA 510(k)		Regulation/Guidance: EU 2016/425 PPE Regulation EU 2017/745 Medical Device Regulation	
	Requirements: ASTM F2100	Test Methods	Requirements: EN 14,683 + AC	Test Methods
Sub-micron particulates filtration efficiency, 0.1 µm (%)	Level 1 ≥ 95 Level 2 ≥ 98 Level 3 ≥ 98	ASTM F2299 Challenge with Latex spheres; face velocity 0.5–25 cm/sec; 107–108 particles/m^3^ with dilution; 1–5 min test	NA	NA
Bacterial filtration efficiency, 3 µm (%)	Level 1 ≥ 95 Level 2 ≥ 98 Level 3 ≥ 98	ASTM F2101 Challenge with *Staphylococcus aureus* at 28.3 L/min; 2200 ± 500 particle per test; 2 min test	Type I ≥ 95 Type II ≥ 98 Type IIR ≥ 98	EN 14,683 + AC Annex B Challenge with *Staphylococcus aureus* at 28.3 L/min; 1700–3000 colony forming units per test; 2 min test
Differential pressure, Pa/cm^2^ (mmH_2_O/cm^2^)	Level 1 < 50 (5) Level 2 < 60 (6) Level 3 < 60 (6)	EN 14,683 + AC Annex B Flow rate 8 L/min	Type I < 40 (4) Type II < 40 (4) Type IIR < 60 (6)	EN 14,683 + AC Annex B Flow rate 8 L/min
Synthetic blood/splash resistance, mmHg (KPa)	Level 1 ≥ 80 (11) Level 2 ≥ 120 (16) Level 3 ≥ 160 (21)	ASTM F1862	Type I NA Type II NA Type IIR ≥ 120 (16)	ISO 22609
Flammability	Level 1–3 to be Class 1	16 CFR 1610	NA	NA
Microbial cleanliness (cfu/g)	NA	NA	Type I ≤ 30 Type II ≤ 30 Type IIR ≤ 30	ISO 11737-1
Biocompatibility	NA	FDA recommends following ISO 10993	Pass	ISO 10993-1 ISO 10993-5 ISO 10993-10
Viral filtration efficiency, 3 µm (%)	No standard	Adapted ASTM F2101 using *PhiX174* virus at 28.3 L/min; 1700–2000 plaque forming units per test	NA	NA

## Data Availability

Not applicable.

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
