# Peer review of "What We Are Learning from COVID-19 for Respiratory Protection: Contemporary and Emerging Issues"

_polymers, 2021, doi:10.3390/polym13234165_

Round 1
Reviewer 1 Report
The manuscript overviewing broad aspects of RPDs with the taste of Covid has been reviewed.
The authors have reviewed the research in-depth enough to analyze and summarize the current research deficiencies and possible future research directions. Therefore, the manuscript covers many aspects of respiratory protection. It would have been nice if the authors had selected one or two key aspects and written their review accordingly.
The keywords are not correctly chosen according to the study. They seem to be misleading the readers.
Check subscripts carefully; for example, refer to lines 1020 and 1034 for H2O2 or 269 for CO2.
The authors have discussed some well-known biocidal methods in the manuscript. Reducing the harmful effect on human health by keeping its usual microbial activities may be worthy of research in this context.
Recently a new technology has been introduced that can kill viruses by the strong positive charge on the surface; I could not find that aspect of the discussion in the manuscript.
Check the completeness of the references; please provide complete information of the references so that the readers might find the cited manuscript easily.
The quality of illustrations is nice, yet consider adding HD images in the revised manuscript.
Just for the author's information, the author might refer to https://doi.org/10.1155/2015/104142; this manuscript has reviewed the technologies for indoor decontamination using textiles. I hope the authors will find it useful.
In the recommendation section, the authors might add/propose a specific design or approach as a solution to overcome the issue raised by the authors.
Author Response
Dear editors and reviewers,
We thank you for spending time on processing and reviewing our manuscript. We appreciate your comments and suggestions, which greatly helped us improve the scientific value and the quality of this manuscript. We have addressed all reviewers’ comments as indicated below. All the changes are tracked in this revision.
We thank reviewer 1 for their comments. We agree that a more focused review on one or two critical issues related to respiratory protection may allow us to dig deeper into the topic and provide a more systematic understanding. However, the purpose of this narrative review was to reveal the emerging issues that have not been understood thoroughly from a technical perspective and to inspire the audience with future directions of respiratory protection development. Thus this review is intended to cover a broad spectrum of issues related to RPDs. We plan to conduct other systematic reviews/meta-analysis in future efforts to offer quantitative and comparable information on some key issues revealed in this article. We have modified the keywords to reflect the key issues revealed and discussed in this article as reviewer suggested, tracked change shown in lines 31-32. We have thoroughly checked the superscripts and subscripts and made changes accordingly, tracked changes of format shown in lines 524-563. In section 6.3.2 we already discussed the positively charged materials (i.e., quaternary ammonium salts) for inactivating the microbes on contact. This section is from lines 1190 to 1203. We have updated the reference list and replaced preprint articles with later published journal articles if available. We have checked all the Url links to make sure they all work. We have updated the figures in this article with high resolution figures also updated as supplemental materials. Photocatalytic oxidation through nanocoating on textiles to provide indoor decontamination is mentioned and cited in the nanomaterial section from line 1144. More discussion of an envisioned next generation RPD is added to the future direction section, from line 1370.
Thanks,
Guowen
Reviewer 2 Report
1) Authors should concentrate more on normal people who are living outside home or streets, effects and controls
2) Rapid changes in social and occupational environments for poor peoples effects and control
3) Asymptomatic and pre-symptomatic details from the beginning to till date
4) RPDs have substantially improved over the past 100 years, So is this the main contribution to control COVID or then why not this effects in the whole world
5) Should explain more on emergence of pathogens with novel transmission and variant strains
Author Response
Dear editors and reviewers,
We thank you for spending time on processing and reviewing our manuscript. We appreciate your comments and suggestions, which greatly helped us improve the scientific value and the quality of this manuscript. We have addressed all reviewers’ comments as indicated below. All the changes are tracked in this revision.
We thank reviewer 2 for their suggestions on the adding/refining of the contents related to RPD user specifics, equality of RPD distributions, evolution of the virus variants and asymptomatic and pre-symptomatic transmission, which are all important in the context of utilizing RPD to protect people from getting COVID-19. Although RPDs have gained great advancement over the past 100 years and are playing a critical role in restricting the spreading of COVID-19, their deficiencies as revealed in this pandemic demonstrate that they are not suitable to be used extensively by occupational workers or to be used daily by the general public. We acknowledge there are still many issues related to RPD usage that are not covered by this article because of our limited knowledge in those fields. For example, the evolution of the virus variants may be better explained by microbiologists and virologists, while the elaboration of the equality of RPD distributions, especially to minority users and economically disadvantaged users, may need expertise from political science, social justice, and culture psychology. As material scientists, mechanical engineers, and PPE designers, we intend to structure this review from a more technical perspective to direct a pathway to better future RPDs. Hence, this article may not have in-depth discussions of the topics raised by the reviewer. However, we do hope other scholars will be inspired by this article, even by its incompleteness, to spur research efforts on the political, social, and cultural aspects of RPD manufacturing, distributions, and usage. We have added some more information related to asymptomatic and pre-symptomatic transmission, and the vaccine's inability to completely stop COVID to stress the importance of universal adoption of RPDs. Added contents can be found in lines 46-64.
Thanks,
Guowen
Reviewer 3 Report
(1) This is a review paper and the content is comprehensive.
(2) I suggest authors to give their critical comments on this topic in their manuscript.
Author Response
Dear editors and reviewers,
We thank you for spending time on processing and reviewing our manuscript. We appreciate your comments and suggestions, which greatly helped us improve the scientific value and the quality of this manuscript. We have addressed all reviewers’ comments as indicated below. All the changes are tracked in this revision.
We thank reviewer 3 for their suggestions that we should add our critical comments on the topics we discussed. In the current version, we already provided some critical comments, such as in line 358-365, 1095-1107. We have added more comments at several places in the article such as 171-174, 613-615, 1370-1379.
Thanks,
Guowen
Reviewer 4 Report
On request of Polymers, I have revised the manuscript entitled “What we are learning from COVID-19 for respiratory protection: Contemporary and emerging issues" by Rui Li and co-authors.
In this review the authors summarized existing knowledge and understanding on respiratory infectious diseases and their protection, discussing the emerging issues that influence the resulting protective and comfort performance of the RPDs, and provide insights in the identified knowledge gaps and future directions with diverse perspectives.
Report
The proposed manuscript addresses an important health issue and a current hot topic. New approaches and a better understanding of COVID19 prevention are much needed.
Therefore, I recommend publishing without any additional changes for the following reasons:
- The proposed manuscript is very well structured and written;
-
The text is clear and easy to read;
- The study is well documented (a bibliography that includes almost 20 pages of references is impressive even for a review)
- The hypothesis is new although not a very original one, but this should not decrease its significance and applicability.
- The problem addressed is significant and challenging.
Author Response
Dear editors and reviewers,
We thank you for spending time on processing and reviewing our manuscript. We appreciate your comments and suggestions, which greatly helped us improve the scientific value and the quality of this manuscript. We have addressed all reviewers’ comments as indicated below. All the changes are tracked in this revision.
We thank reviewer 4 for their comments on this article. We hope this article will inspire the audience, evoke integrated transdisciplinary efforts in enhancing the understanding of respiratory protection principles and technologies, and promote the development of next generation RPDs.
Thanks,
Guowen